# C-Mixup: Improving Generalization in Regression

**Huaxiu Yao**[1*]**, Yiping Wang**[2*]**, Linjun Zhang**[3]**, James Zou**[1]**, Chelsea Finn**[1]

[1]Stanford University, [2]Zhejiang University, [3]Rutgers University
[1]{huaxiu,cbfinn}@cs.stanford.edu, jamesz@stanford.edu
[2]yipingwang6161@gmail.com, [3]linjun.zhang@rutgers.edu

## Abstract

Improving the generalization of deep networks is an important open challenge, particularly in domains without plentiful data. The mixup algorithm improves generalization by linearly interpolating a pair of examples and their corresponding labels. These interpolated examples augment the original training set. Mixup has shown promising results in various classification tasks, but systematic analysis of mixup in regression remains underexplored. Using mixup directly on regression labels can result in arbitrarily incorrect labels. In this paper, we propose a simple yet powerful algorithm, C-Mixup, to improve generalization on regression tasks. In contrast with vanilla mixup, which picks training examples for mixing with uniform probability, C-Mixup adjusts the sampling probability based on the similarity of the labels. Our theoretical analysis confirms that C-Mixup with label similarity obtains a smaller mean square error in supervised regression and meta-regression than vanilla mixup and using feature similarity. Another benefit of C-Mixup is that it can improve out-of-distribution robustness, where the test distribution is different from the training distribution. By selectively interpolating examples with similar labels, it mitigates the effects of domain-associated information and yields domain-invariant representations. We evaluate C-Mixup on eleven datasets, ranging from tabular to video data. Compared to the best prior approach, C-Mixup achieves 6.56%, 4.76%, 5.82% improvements in in-distribution generalization, task generalization, and out-of-distribution robustness, respectively. Code is released at https://github.com/huaxiuyao/C-Mixup.

## 1 Introduction

Deep learning practitioners commonly face the challenge of overfitting. To improve generalization, prior works have proposed a number of techniques, including data augmentation [3, 10, 12, 81, 82] and explicit regularization [15, 38, 60]. Representatively, mixup [82, 83] densifies the data distribution and implicitly regularizes the model by linearly interpolating the features of randomly sampled pairs of examples and applying the same interpolation on the corresponding labels. Despite mixup having demonstrated promising results in improving generalization in classification problems, it has rarely been studied in the context of regression with continuous labels, on which we focus in this paper.

In contrast to classification, which formalizes the label as a one-hot vector, the goal of regression is to predict a continuous label from each input. Directly applying mixup to input features and labels in regression tasks may yield arbitrarily incorrect labels. For example, as shown in Figure 1(a), ShapeNet1D pose prediction [18] aims to predict the current orientation of the object relative to its canonical orientation. We randomly select three mixing pairs and show the mixed images and labels in Figure 1(b), where only pair 1 exhibits reasonable mixing results. We thus see that sampling mixing pairs uniformly from the dataset introduces a number of noisy pairs.

---

[*]Equal contribution. This work was done when Yiping Wang was remotely co-mentored by Huaxiu Yao and Linjun Zhang.

36th Conference on Neural Information Processing Systems (NeurIPS 2022).

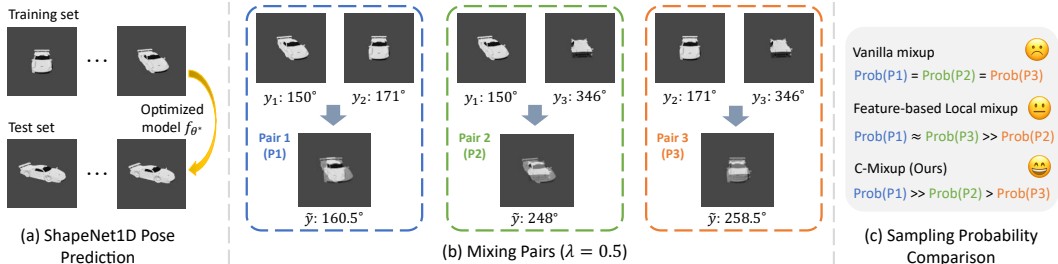

Figure 1: Illustration of C-Mixup on ShapeNet1D pose prediction. $\lambda$ represents the interpolation ratio. (a) ShapeNet1D pose prediction task, aiming to predict the current orientation of the object relative to its canonical orientation. (b) Three mixing pairs are randomly picked, where the interpolated images are visualized and $\tilde{y}$ represents interpolated labels. (c) Illustration of a rough comparison of sampling probabilities among three mixing pairs in (b). The Euclidean distance measures input feature distance and the corresponding results between examples in pairs 1, 2, 3 are $1.51 \times 10^5$, $1.82 \times 10^5$, $1.50 \times 10^5$, respectively. Hence, pairs 1 and 3 have similar results, leading to similar sampling probabilities. C-Mixup is able to assign higher sampling probability for more reasonable mixing pairs.

In this paper, we aim to adjust the sampling probability of mixing pairs according to the similarity of examples, resulting in a simple training technique named **C-Mixup**. Specifically, we employ a Gaussian kernel to calculate the sampling probability of drawing another example for mixing, where closer examples are more likely to be sampled. Here, the core question is: *how to measure the similarity between two examples?* The most straightforward solution is to compute input feature similarity. Yet, using input similarity has two major downsides when dealing with high-dimensional data such as images or time-series: substantial computational costs and lack of good distance metrics. Specifically, it takes considerable time to compute pairwise similarities across all samples, and directly applying classical distance metrics (e.g., Euclidean distance, cosine distance) does not reflect the high-level relation between input features. In the ShapeNet1D rotation prediction example (Figure 1(a)), pair 1 and 3 have close input similarities, while only pair 1 can be reasonably interpolated.

To overcome these drawbacks, C-Mixup instead uses label similarity, which is typically much faster to compute since the label space is usually low dimensional. In addition to the computational advantages, C-Mixup benefits three kinds of regression problems. *First*, it empirically improves in-distribution generalization in supervised regression compared to using vanilla mixup or using feature similarity. *Second*, we extend C-Mixup to gradient-based meta-learning by incorporating it into MetaMix, a mixup-based task augmentation method [74]. Compared to vanilla MetaMix, C-Mixup empirically improves task generalization. *Third*, C-Mixup is well-suited for improving out-of-distribution robustness without domain information, particularly to covariate shift (see the corresponding example in Appendix A.1). By performing mixup on examples with close continuous labels, examples with different domains are mixed. In this way, C-Mixup encourages the model to rely on domain-invariant features to make prediction and ignore unrelated or spurious correlations, making the model more robust to covariate shift.

The primary contribution of this paper is C-Mixup, a simple and scalable algorithm for improving generalization in regression problems. In linear or monotonic non-linear models, our theoretical analysis shows that C-Mixup improves generalization in multiple settings compared to vanilla mixup or compared to using feature similarities. Moreover, our experiments thoroughly evaluate C-Mixup on eleven datasets, including many large-scale real-world applications like drug-target interaction prediction [28], ejection fraction estimation with echocardiogram videos [50], poverty estimation with satellite imagery [78]. Compared to the best prior method, the results demonstrate the promise of C-Mixup with 6.56%, 4.76%, 5.82% improvements in in-distribution generalization, task generalization, and out-of-distribution robustness, respectively.

## 2    Preliminaries

In this section, we define notation and describe the background of ERM and mixup in the supervised learning setting, and MetaMix in the meta-learning setting for task generalization.

**ERM.** Assume a machine learning model $f$ with parameter space $\Theta$. In this paper, we consider the setting where one predicts the continuous label $y \in \mathcal{Y}$ according to the input feature $x \in \mathcal{X}$. Given a loss function $\ell$, we train a model $f_\theta$ under the empirical training distribution $P^{tr}$ with the following objective, and get the optimized parameter $\theta^* \in \Theta$:

$$\theta^* \leftarrow \arg\min_{\theta \in \Theta} \mathbb{E}_{(x,y) \sim P^{tr}}[\ell(f_\theta(x), y)]. \tag{1}$$

Typically, we expect the model to perform well on unseen examples drawn from the test distribution $P^{ts}$. We are interested in both *in-distribution* ($P^{tr} = P^{ts}$) and *out-of-distribution* ($P^{tr} \neq P^{ts}$) settings.

**Mixup.** The mixup algorithm samples a pair of instances $(x_i, y_i)$ and $(x_j, y_j)$, sampled uniformly at random from the training dataset, and generates new examples by performing linear interpolation on the input features and corresponding labels as:

$$\tilde{x} = \lambda \cdot x_i + (1 - \lambda) \cdot x_j, \ \tilde{y} = \lambda \cdot y_i + (1 - \lambda) \cdot y_j, \tag{2}$$

where the interpolation ratio $\lambda \in [0, 1]$ is drawn from a Beta distribution, i.e., $\lambda \sim \text{Beta}(\alpha, \alpha)$. The interpolated examples are then used to optimize the model as follows:

$$\theta^* \leftarrow \arg\min_{\theta \in \Theta} \mathbb{E}_{(x_i, y_i),(x_j, y_j) \sim P^{tr}}[\ell(f_\theta(\tilde{x}), \tilde{y})]. \tag{3}$$

**Task Generalization and MetaMix.** In this paper, we also investigate few-shot *task generalization* under the gradient-based meta-regression setting. Given a task distribution $p(\mathcal{T})$, we assume each task $\mathcal{T}_m$ is sampled from $p(\mathcal{T})$ and is associated with a dataset $\mathcal{D}_m$. A support set $\mathcal{D}_m^s = \{(X_m^s, Y_m^s)\} = \{(x_{m,i}^s, y_{m,i}^s)\}_{i=1}^{N^s}$ and a query set $\mathcal{D}_m^q = \{(X_m^s, Y_m^s)\} = \{(x_{m,j}^q, y_{m,j}^q)\}_{i=1}^{N^q}$ are sampled from $\mathcal{D}_m$. Representatively, in model-agnostic meta-learning (MAML) [14], given a predictive model $f$ with parameter $\theta$, it aims to learn an initialization $\theta^*$ from meta-training tasks $\{\mathcal{T}_m\}_{m=1}^{|M|}$. Specifically, at the meta-training phase, MAML obtained the task-specific parameter $\phi_m$ for each task $\mathcal{T}_m$ by performing a few gradient steps starting from $\theta$. Then, the corresponding query set $\mathcal{D}_m^q$ is used to evaluate the performance of the task-specific model and optimize the model initialization as:

$$\theta^* := \arg\min_\theta \frac{1}{|M|} \sum_{i=1}^{|M|} \mathcal{L}(f_{\phi_m}; \mathcal{D}_m^q), \ where \ \phi_m = \theta - \alpha \nabla_\theta \mathcal{L}(f_\theta; \mathcal{D}_m^s) \tag{4}$$

At the meta-testing phase, for each meta-testing task $\mathcal{T}_t$, MAML fine-tunes the learned initialization $\theta^*$ on the support set $\mathcal{D}_t^s$ and evaluates the performance on the corresponding query set $\mathcal{D}_t^q$.

To improve task generalization, MetaMix [74] adapts mixup (Eqn. (3)) to meta-learning, which linearly interpolates the support set and query set and uses the interpolated set to replace the original query set $\mathcal{D}_m^q$ in Eqn. (4). Specifically, the interpolated query set is formulated as:

$$\tilde{X}_m^q = \lambda X_m^s + (1 - \lambda)X_m^q, \ \tilde{Y}_m^q = \lambda Y_m^s + (1 - \lambda)Y_m^q, \tag{5}$$

where $\lambda \sim \text{Beta}(\alpha, \alpha)$.

## 3  Mixup for Regression (C-Mixup)

For continuous labels, the example in Figure 1(b) illustrates that applying vanilla mixup to the entire distribution is likely to produce arbitrary labels. To resolve this issue, C-Mixup proposes to sample closer pairs of examples with higher probability. Specifically, given an example $(x_i, y_i)$, C-Mixup introduces a symmetric Gaussian kernel to calculate the sampling probability $P((x_j, y_j)|(x_i, y_i))$ for another $(x_j, y_j)$ example to be mixed as follows:

$$P((x_j, y_j)|(x_i, y_i)) \propto \exp\left(-\frac{d(i, j)}{2\sigma^2}\right) \tag{6}$$

where $d(i, j)$ represents the distance between the examples $(x_i, y_i)$ and $(x_j, y_j)$, and $\sigma$ describes the bandwidth. For the example $(x_i, y_i)$, the set $\{P((x_j, y_j)|(x_i, y_i))|\forall j\}$ is then normalized to a probability mass function that sums to one.

---

**Algorithm 1** Training with C-Mixup

---

**Require:** Learning rates $\eta$; Shape parameter $\alpha$
**Require:** Training data $\mathcal{D} := \{(x_i, y_i)\}_{i=1}^{N}$
 1: Randomly initialize model parameters $\theta$
 2: Calculate pairwise distance matrix $P$ via Eqn. (6)
 3: **while** not converge **do**
 4:     Sample a batch of examples $\mathcal{B} \sim \mathcal{D}$
 5:     **for** each example $(x_i, y_i) \in \mathcal{B}$ **do**
 6:         Sample $(x_j, y_j)$ from $P(\cdot \mid (x_i, y_i))$ and $\lambda$ from $\text{Beta}(\alpha, \alpha)$
 7:         Interpolate $(x_i, y_i), (x_j, y_j)$ to get $(\tilde{x}, \tilde{y})$ according to Eqn. (2)
 8:     Use interpolated examples to update the model via Eqn. (3)

---

One natural way to compute the distance is using the input feature $x$, i.e., $d(i, j) = d(x_i, x_j)$. However, when dealing with the high-dimensional data such as images or videos, we lack good distance metrics to capture structured feature information and the distances can be easily influenced by feature noise. Additionally, computing feature distances for high-dimensional data is time-consuming. Instead, C-Mixup leverages the labels with $d(i, j) = d(y_i, y_j) = \|y_i - y_j\|_2^2$, where $y_i$ and $y_j$ are vectors with continuous values. The dimension of label is typically much smaller than that of the input feature, therefore reducing computational costs (see more discussions about compuatational efficiency in Appendix A.3). The overall algorithm of C-Mixup is described in Alg. 1 and we detail the difference between C-Mixup and mixup in Appendix A.4. According to Alg. 1, C-Mixup assigns higher probabilities to example pairs with closer continuous labels. In addition to its computational benefits, C-Mixup improves generalization on three distinct kinds of regression problems – in-distribution generalization, task generalization, and out-of-distribution robustness, which is theoretically and empirically justified in the following sections.

## 4 Theoretical Analysis

In this section, we theoretically explain how C-Mixup benefits in-distribution generalization, task generalization, and out-of-distribution robustness.

### 4.1 C-Mixup for Improving In-Distribution Generalization

In this section, we show that C-Mixup provably improves in-distribution generalization when the features are observed with noise, and the response depends on a small fraction of the features in a monotonic way. Specifically, we consider the following single index model with measurement error,

$$y = g(\theta^\top z) + \epsilon, \tag{7}$$

where $\theta \in \mathbb{R}^p$ and $\epsilon$ is a sub-Gaussian random variable and $g$ is a monotonic transformation. Since images are often inaccurately observed in practice, we assume the feature $z$ is observed or measured with noise, and denote the observed value by $x$: $x = z + \xi$ with $\xi$ being a random vector with mean 0 and covariance matrix $\sigma_\xi^2 I$. We assume $g$ to be monotonic to model the nearly one-to-one correspondence between causal features (e.g., the car pose in Figure 1(a)) and labels (rotation) in the in-distribution setting. The out-of-distribution setting will be discussed in Section 4.3. We would like to also comment that the single index model has been commonly used in econometrics, statistics, and deep learning theory [19, 27, 47, 53, 73].

Suppose we have $\{(x_i, y_i)\}_{i=1}^{N}$ i.i.d. drawn from the above model. We first follow the single index model literature (e.g. [73]) and estimate $\theta$ by minimizing the square error $\sum_{i=1}^{n} (\tilde{y}_i - \tilde{x}_i^\top \theta)^2$, where the $(\tilde{x}_i, \tilde{y}_i)'s$ are the augmented data by either vanilla mixup, mixup with input feature similarity, and C-Mixup. We denote the solution by $\theta_{mixup}^*$, $\theta_{feat}^*$, and $\theta_{C-Mixup}^*$ respectively. Given an estimate $\theta^*$, we estimate $g$ by $\hat{g}$ via the standard nonparametric kernel estimator [64] (we specify this in detail in Appendix B.1 for completeness) using the augmented data. We consider the mean square error metric as $\text{MSE}(\theta) = \mathbb{E}[(y - \hat{g}(\theta^\top x))^2]$, and then have the following theorem (proof: Appendix B.1):

**Theorem 1.** *Suppose $\theta \in \mathbb{R}^p$ is sparse with sparsity $s = o(\min\{p, \sigma_\xi^2\})$, $p = o(N)$ and $g$ is smooth with $c_0 < g' < c_1$, $c_2 < g'' < c_3$ for some universal constants $c_0, c_1, c_2, c_3 > 0$. There exists a*

*distribution on $x$ with a kernel function, such that when the sample size $N$ is sufficiently large, with probability $1 - o(1)$,*

$$\text{MSE}(\theta^*_{C-Mixup}) < \min(\text{MSE}(\theta^*_{feat}), \text{MSE}(\theta^*_{mixup})). \qquad (8)$$

The high-level intuition of why C-Mixup helps is that the vanilla mixup imposes linearity regularization on the relationship between the feature and response. When the relationship is strongly nonlinear and one-to-one, such a regularization hurts the generalization, but could be mitigated by C-Mixup.

## 4.2 C-Mixup for Improving Task Generalization

The second benefit of C-Mixup is improving task generalization in meta-learning when the data from each task follows the model discussed in the last section. Concretely, we apply C-Mixup to MetaMix [74]. For each query example, the support example with a more similar label will have a higher probability of being mixed. The algorithm of C-Mixup on MetaMix is summarized in Appendix A.2.

Similar to in-distribution generalization analysis, we consider the following data generative model: for the $m$-th task ($m \in [M]$), we have $(x^{(m)}, y^{(m)}) \sim \mathcal{T}_m$ with

$$y^{(m)} = g_m(\theta^\top z^{(m)}) + \epsilon \quad \text{and} \quad x^{(m)} = z^{(m)} + \epsilon^{(m)}. \qquad (9)$$

Here, $\theta$ denotes the globally-shared representation, and $g_m$'s are the task-specific transformations. Note that, the formulation is close to [54] and wildly applied to theoretical analysis of meta-learning [74, 77]. Following similar spirit of [62] and last section, we obtain the estimation of $\theta$ by $\theta^* = \frac{1}{M} \sum_{m=1}^{M} (\mathbb{E}_{(\tilde{x}^{(m)}, \tilde{y}^{(m)}) \in \hat{\mathcal{D}}_m} [\tilde{y}^{(m)} - \theta^\top \tilde{x}^{(m)}])$. Here, $\hat{\mathcal{D}}_m$ denotes the generic dataset augmented by different approaches, including the vanilla MetaMix, MetaMix with input feature similarity, and C-Mixup. We denote these approaches by $\theta^*_{MetaMix}$, $\theta^*_{Meta-feat}$, and $\theta^*_{Meta-C-Mixup}$ respectively. For a new task $\mathcal{T}_t$, we again use the standard nonparametric kernel estimator to estimate $g_t$ via the augmented target data. We then consider the following error metric $\text{MSE}_{\text{Target}}(\theta^*) = \mathbb{E}_{(x,y) \sim \mathcal{T}_t}[(y - \hat{g}_t(\theta^{*\top} x))^2]$.

Based on this metric, we get the following theorem to show the promise of C-Mixup in improving task generalization (see Appendix B.2 for detailed proof). Here, C-Mixup achieves smaller $\text{MSE}_{\text{Target}}$ compared to vanilla MetaMix and MetaMix with input feature similarity.

**Theorem 2.** *Let $N = \sum_{m=1}^{M} N_m$ and $N_m$ is the number of examples of $\mathcal{T}_m$. Suppose $\theta_k$ is sparse with sparsity $s = o(\min\{d, \sigma_\xi^2\})$, $p = o(N)$ and $g_m$'s are smooth with $0 < g'_m < c_1$, $c_2 < g''_m < c_3$ for some universal constants $c_1, c_2, c_3 > 0$ and $m \in [M] \cup \{t\}$. There exists a distribution on $x$ with a kernel function, such that when the sample size $N$ is sufficiently large, with probability $1 - o(1)$,*

$$\text{MSE}_{\text{Target}}(\theta^*_{Meta-C-Mixup}) < \min(\text{MSE}_{\text{Target}}(\theta^*_{Meta-feat}), \text{MSE}_{\text{Target}}(\theta^*_{MetaMix})). \qquad (10)$$

## 4.3 C-Mixup for Improving Out-of-distribution Robustness

Finally, we show that C-Mixup improves OOD robustness in the covariate shift setting where some unrelated features vary across different domains. In this setting, we regard the entire data distribution consisting of $\mathcal{E} = \{1, \ldots, E\}$ domains, where each domain is associated with a data distribution $P_e$ for $e \in \mathcal{E}$. Given a set of training domains $\mathcal{E}^{tr} \subseteq \mathcal{E}$, we aim to make the trained model generalize well to an unseen test domain $\mathcal{E}^{ts}$ that is not necessarily in $\mathcal{E}^{tr}$. Here, we focus on covariate shift, i.e., the change of $P_e$ among domains is only caused by the change of marginal distribution $P_e(X)$, while the conditional distribution $P_e(Y|X)$ is fixed across different domains.

To overcome covariate shift, mixing examples with close labels without considering domain information can effectively average out domain-changeable correlations and make the predicted values rely on the invariant causal features. To further understand how C-Mixup improves the robustness to covariate shift, we provide the following theoretical analysis.

We assume the training data $(x_i, y_i)_{i=1}^n$ follows $x_i = (z_i; a_i) \in \mathbb{R}^{p_1+p_2}$ and $y_i = \theta^\top x_i + \epsilon_i$, where $z_i \in \mathbb{R}^{p_1}$ and $a_i \in \mathbb{R}^{p_2}$ are regarded as invariant and domain-changeable unrelated features, respectively, and the last $p_2$ coordinates of $\theta \in \mathbb{R}^{p_1+p_2}$ are 0. Now we consider the case where the training data consists of a pair of domains with almost identical invariant features and opposite domain-changeable features, i.e., $x_i = (z_i, a_i)$, $x'_i = (z'_i, a'_i)$, where $z_i \sim \mathcal{N}_{p_1}(0, \sigma_x^2 I_{p_1})$, $z'_i = z_i + \epsilon'_i$, $a_i \sim \mathcal{N}_{p_2}(0, \sigma_a^2 I_{p_2})$, $a'_i = -a_i + \epsilon''_i$. $\epsilon_i, \epsilon'_i, \epsilon_i''$ are noise terms with mean 0 and sub-Gaussian norm bounded by $\sigma_\epsilon$. We use ridge estimator $\theta^*(k) = \arg\min_\theta (\sum_i \|y_i - \theta^\top x_i\|^2 + k\|\theta\|^2)$ to reflect the

implicit regularization effect of deep neural networks [48]. The covariant shift happens when the test domains have different $a_i$ distributions compared with training domains. We then have the following theorem to show that C-Mixup can improve robustness to covariant shift (see proof in Appendix B.3):

**Theorem 3.** *Supposed for some* $\max(\exp(-n^{1-o(1)}), \exp(-\frac{p_1^2}{2n})) < \delta \ll 1$, *we have variance constraints:* $\sigma_a = c_1\sigma_x$, $\sigma_x \geq c_2 \max(\frac{n^{5/2}}{\|\theta\|\delta}\sigma_\epsilon, \frac{\sqrt{p_2}\|\theta\|}{\sqrt{n}p_1})$ *and* $\sigma_\epsilon^2 \leq \frac{c_3}{pn^{3/2}}$ . *Then for any penalty k satisfies* $c_4\sqrt{\frac{p_2}{p_1}}n^{1/4+o(1)} < k < c_5\min(\frac{\sigma_x}{\|\theta\|}\sqrt{p_1n^{1-o(1)}}, n)$ *and bandwidth h satisfies* $0 < h \leq c_6\frac{l}{\sqrt{\log(n^2/p_1)}}$ *in C-Mixup, when $n$ is sufficiently large, with probability at least $1 - o(1)$, we have*

$$\text{MSE}(\theta^*_{C-Mixup}) < \min(\text{MSE}(\theta^*_{feat}), \text{MSE}(\theta^*_{mixup})), \tag{11}$$

*where $c_1 \geq 1$, $c_2$, $c_3$, $c_4$, $c_5$, $c_6 > 0$ are universal constants, $l = \min_{i \neq j}|y_i - y'_j|$ and $p_1 \ll n < p_1^2$.*

## 5 Experiments

In this section, we evaluate the performance of C-Mixup, aiming to answer the following questions: **Q1:** Compared to corresponding prior approaches, can C-Mixup improve the in-distribution, task generalization, and out-of-distribution robustness on regression? **Q2:** How does C-Mixup perform compared to using other distance metrics? **Q3:** Is C-Mixup sensitive to the choice of bandwidth $\sigma$ in the Gaussian kernel of Eqn. (6)? In our experiments, we apply cross-validation to tune all hyperparameters with grid search.

### 5.1 In-Distribution Generalization

**Datasets.** We use the following five datasets to evaluate the performance of in-distribution generalization (see Appendix C.1 for detailed data statistics). **(1)&(2) Airfoil Self-Noise (Airfoil) and NO2 [35]** are both are tabular datasets, where airfoil contains aerodynamic and acoustic test results of airfoil blade sections and NO2 aims to predict the mount of air pollution at a particular location. **(3)&(4): Exchange-Rate, and Electricity [40]** are two time-series datasets, where Exchange-Rate reports the collection of the daily exchange rates and Electricity is used to predict the hourly electricity consumption. **(5) Echocardiogram Videos (Echo) [50]** is a ejection fraction prediction dataset, which consists of a series of videos illustrating the heart from different aspects.

**Comparisons and Experimental Setups.** We compare C-Mixup with mixup and its variants (Manifold mixup [68], k-Mixup [20] and Local Mixup [5]) that can be easily to adapted to regression tasks. We also compare to MixRL, a recent reinforcement learning framework to select mixup pairs in regression. Note that, for k-Mixup, Local Mixup, MixRL, and C-Mixup, we apply them to both mixup and Manifold Mixup and report the best-performing combination.

For Airfoil and NO2, we employ a three-layer fully connected network as the backbone model. We use LST-Attn [40] for the Exchange-Rate and Electricity, and EchoNet-Dynamic [50] for predicting the ejection fraction. We use Root Mean Square Error (RMSE) and Mean Averaged Percentage Error (MAPE) as evaluation metrics. Detailed experimental setups are in Appendix C.2.

**Results.** We report the results in Table 1 and have the following observations. *First*, vanilla mixup and manifold mixup are typically less performant than ERM when they are applied directly to regression tasks. These results support our hypothesis that random selection of example pairs for mixing may produce arbitrary inaccurate virtual labels. *Second*, while limiting the scope of interpolation (e.g., k-Mixup, Manifold k-Mixup, Local Mixup) helps to improve generalization in most cases, the performance of these

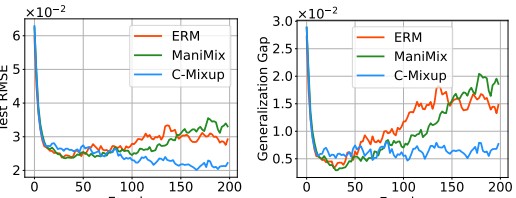

Figure 2: Overfitting Analysis on Exchange-Rate. C-Mixup achieves better test performance and smaller generalization gap.

approaches is inconsistent across different datasets. As an example, Local Mixup outperforms ERM in NO2 and Exchange-Rate, but fails to benefit results in Electricity. *Third*, even in more complicated datasets, such as Electricity and Echo, MixRL performs worse than Mixup and ERM, indicating that it is non-trivial to train a policy network that works well. *Finally*, C-Mixup consistently outperforms mixup and its variants, ERM, and MixRL, demonstrating its capability to improve in-distribution generalization on regression problems.

Table 1: Results for in-distribution generalization. We report the average RMSE and MAPE of three seeds. Full results with standard deviation are reported in Appendix C.4. The best results and second best results are **bold** and underlined, respectively.

|  | Airfoil | | NO2 | | Exchange-Rate | | Electricity | | Echo | |
|---|---|---|---|---|---|---|---|---|---|---|
|  | RMSE | MAPE | RMSE | MAPE | RMSE | MAPE | RMSE | MAPE | RMSE | MAPE |
| ERM | 2.901 | 1.753% | 0.537 | 13.615% | 0.0236 | 2.423% | 0.0581 | 13.861% | 5.402 | 8.700% |
| mixup | 3.730 | 2.327% | 0.528 | 13.534% | 0.0239 | 2.441% | 0.0585 | 14.306% | 5.393 | 8.838% |
| Mani mixup | 3.063 | 1.842% | 0.522 | 13.382% | 0.0242 | 2.475% | 0.0583 | 14.556% | 5.482 | 8.955% |
| k-Mixup | 2.938 | 1.769% | 0.519 | 13.173% | 0.0236 | 2.403% | 0.0575 | 14.134% | 5.518 | 9.206% |
| Local Mixup | 3.703 | 2.290% | 0.517 | 13.202% | 0.0236 | 2.341% | 0.0582 | 14.245% | 5.652 | 9.313% |
| MixRL | 3.614 | 2.163% | 0.527 | 13.298% | 0.0238 | 2.397% | 0.0585 | 14.417% | 5.618 | 9.165% |
| **C-Mixup (Ours)** | **2.717** | **1.610%** | **0.509** | **12.998%** | **0.0203** | **2.041%** | **0.0570** | **13.372%** | **5.177** | **8.435%** |

**Analysis of Overfitting.** In Figure 2, we visualize the test loss (RMSE) and the generalization gap between training loss and test loss of ERM, Manifold Mixup, and C-Mixup with respect to the training epoch for Exchange-Rate. More results are illustrated in Appendix C.3. Compared with ERM and C-Mixup, the better test performance and smaller generalization gap of C-Mixup further demonstrates its ability to improve in-distribution generalization and mitigate overfitting.

## 5.2 Task Generalization

**Dataset and Experimental Setups.** To evaluate task generalization in meta-learning settings, we use two rotation prediction datasets named as ShapeNet1D [18] and Pascal1D [71, 79], the goal of both datasets is to predict an object's rotation relative to the canonical orientation. Each task is rotation regression for one object, where the model takes a 128×128 grey-scale image as the input, and the output is an azimuth angle normalized between [0, 10]. We detail the description in Appendix D.1.

Since MetaMix outperforms most other methods in PASCAL3D in Yao et al. [74], we only select two other representative approaches – Meta-Aug [55] and MR-MAML [79] for comparison. We further apply k-Mixup and Local Mixup to MetaMix, which are called as k-MetaMix and Local MetaMix, respectively. Following Yin et al. [79], the base model consists of an encoder with three convolutional blocks and a decoder with four convolutional blocks. Hyperparameters are listed in Appendix D.2.

**Results.** Table 2 shows the average MSE with a 95% confidence interval over 6000 meta-testing tasks. Our results corroborate the findings of Yao et al. [74] that MetaMix improves performance compared with non-mixup approaches (i.e., MAML, MR-MAML, Meta-Aug). Similar to our previous in-distribution generalization findings, the better performance of Local MetaMix over MetaMix demonstrates the effectiveness of interpolating nearby examples. By mixing query examples with support examples that have similar labels, C-Mixup outperforms

Table 2: Meta-regression performance (MSE ± 95% confidence interval) on 6000 meta-test tasks.

| Model | ShapeNet1D ↓ | PASCAL3D ↓ |
|---|---|---|
| MAML | 4.698 ± 0.079 | 2.370 ± 0.072 |
| MR-MAML | 4.433 ± 0.083 | 2.276 ± 0.075 |
| Meta-Aug | 4.312 ± 0.086 | 2.298 ± 0.071 |
| MetaMix | 4.275 ± 0.082 | 2.135 ± 0.070 |
| k-MetaMix | 4.268 ± 0.078 | 2.091 ± 0.069 |
| Local MetaMix | 4.201 ± 0.087 | 2.107 ± 0.078 |
| **C-Mixup (Ours)** | **4.024 ± 0.081** | **1.995 ± 0.067** |

all of the other approaches on both datasets, verifying its effectiveness of improving task generalization.

## 5.3 Out-of-Distribution Robustness

**Synthetic Dataset with Subpopulation Shift.** Similar to synthetic datasets with subpopulation shifts in classification, e.g., ColoredMNIST [4], we build a synthetic regression dataset – RCFashion-MNIST (**RCF-MNIST**), which is illustrated in Figure 3. Built on the FashionMNIST [18], the goal of RCF-MNIST is to predict the angle of rotation for each object. As shown in Figure 3, we color each image with a color between red and blue. There is a spurious correlation between angle (label) and

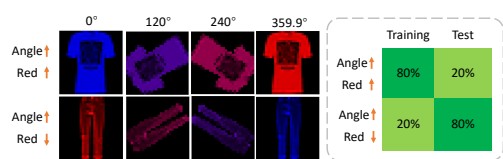

Figure 3: Illustration of RCF-MNIST.

Table 3: Results for out-of-distribution robustness. We report the average and worst-domain (primary metric) performance here and the full results are listed in Appendix E.3. Sub. Shift means Subpopulation Shift. Higher $R$ or lower RMSE represent better performance. mixup and C-Mixup uses the same type of mixup variants reported in Table E.2. For PovertyMap, most results are copied from WILDS benchmark [34] and worst-domain performance is the primary metric. We bold the best results and underline the second best results.

| | Sub. Shift | Domain Shift | | | | | | | |
| | RCF-MNIST | PovertyMap ($R$) | | Crime (RMSE) | | SkillCraft (RMSE) | | DTI ($R$) | |
| | Avg. (RMSE) ↓ | Avg. ↑ | Worst ↑ | Avg. ↓ | Worst ↓ | Avg. ↓ | Worst ↓ | Avg. ↑ | Worst ↑ |
|---|---|---|---|---|---|---|---|---|---|
| ERM | 0.162 | 0.80 | 0.50 | 0.134 | 0.173 | 5.887 | 10.182 | 0.464 | 0.429 |
| IRM | 0.153 | 0.77 | 0.43 | 0.127 | 0.155 | 5.937 | 7.849 | 0.478 | 0.432 |
| IB-IRM | 0.167 | 0.78 | 0.40 | 0.127 | 0.153 | 6.055 | 7.650 | 0.479 | 0.435 |
| V-REx | 0.154 | **0.83** | 0.48 | 0.129 | 0.157 | 6.059 | 7.444 | 0.485 | 0.435 |
| CORAL | 0.163 | 0.78 | 0.44 | 0.133 | 0.166 | 6.353 | 8.272 | 0.483 | 0.432 |
| GroupDRO | 0.232 | 0.75 | 0.39 | 0.138 | 0.168 | 6.155 | 8.131 | 0.442 | 0.407 |
| Fish | 0.263 | 0.80 | 0.30 | 0.128 | 0.152 | 6.356 | 8.676 | 0.470 | 0.443 |
| mixup | 0.176 | 0.81 | 0.46 | 0.128 | 0.154 | 5.764 | 9.206 | 0.465 | 0.437 |
| **C-Mixup (Ours)** | **0.146** | 0.81 | **0.53** | **0.123** | **0.146** | **5.201** | **7.362** | **0.498** | **0.458** |

color in training set. The larger the angle, the more red the color. In test set, we reverse spurious correlations to simulate distribution shift. We provide detailed description in Appendix E.1.

**Real-world Datasets with Domain Shifts.** In the following, we briefly discuss the real-world datasets with domain shifts; the detailed data descriptions in Appendix E.1: **(1) PovertyMap [34]** is a satellite image regression dataset, aiming to estimate asset wealth in countries that are not shown in the training set. **(2) Communities and Crime (Crime)** [56] is a tabular dataset, where the problem is to predict total number of violent crimes per 100K population and we aim to generalize the model to unseen states. **(3) SkillCraft1 Master Table (SkillCraft)** [6] is a tabular dataset, aiming to predict the mean latency from the onset of a perception action cycles to their first action in milliseconds. Here, "LeagueIndex" is treated as domain information. **(4) Drug-target Interactions (DTI)** [28] is aiming to predict out-of-distribution drug-target interactions in 2019-2020 after training on 2013-2018.

**Comparisons and Experimental Setups.** To evaluate the out-of-distribution robustness of C-Mixup, we compare it with nine invariant learning approaches that can be adapted to regression tasks, including ERM, IRM [4], IB-IRM [1], V-REx [39], CORAL [41], DRNN [16], GroupDRO [58], Fish [59], and mixup. All approaches use the same backbone model, where we adopt ResNet-50, three-layer full connected network, and DeepDTA [51] for Poverty, Crime and SkillCraft, and DTI, respectively. Following the original papers of PovertyMap [78] and DTI [28], we use $R$ value to evaluate the performance. For RCF-MNIST, Crime and SkillCraft, we use RMSE as the evaluation metric. In datasets with domain shifts, we report both average and worst-domain (primary metric for PovertyMap [34]) performance. Detailed hyperparameters are listed in Appendix E.2.

**Results.** We report both average and worst-domain performance in Table 3. According to the results, we can see that the performance of prior invariant learning approaches is not stable across datasets. For example, IRM and CORAL outperform ERM on Camelyon17, but fail to improve the performance on PovertyMap. C-Mixup instead consistently shows the best performance regardless of the data types, indicating its effeacy in improving robustness to covariate shift.

**Analysis of Learned Invariance.** Following Yao et al. [76], we analyze the domain invariance of the model learned by C-Mixup. For regression, we measure domain invariance as pairwise divergence of the last hidden representation. Since the labels are continuous, we first evenly split examples into $\mathcal{C} = \{1, \ldots, C\}$ bins according to their labels, where $C = 10$. After splitting, we perform kernel density estimation to estimate the probability density function $P(h_e^c)$ of the last hidden representation. We calculate the pairwise divergence as $\text{Inv} = \frac{1}{|\mathcal{C}||\mathcal{E}|^2} \sum_{c \in \mathcal{C}} \sum_{e', e \in \mathcal{E}} \text{KL}(P(g_E^c \mid E = e) | P(g_E^c \mid E = e'))$. The results on Crime and SkillCraft are reported in Figure 4, where smaller number denotes stronger invariance.

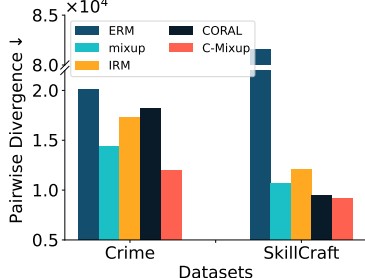

Figure 4: Results of invariance analysis. Smaller pairwise divergence value represents stronger invariance.

We observe that C-Mixup learns better invariant representation compared to prior invariant learning approaches.

## 5.4 Analysis of C-Mixup

In this section, we conduct three analyses including the compatibility of C-Mixup, alternative distance metrics, and sensitivity of bandwidth. In Appendix F.3.2 and F.4, we analyze the sensitivity of hyperparameter $\alpha$ in Beta distribution and the robustness of C-Mixup to label noise, respectively.

**I. Compatibility of C-Mixup.** C-Mixup is a complementary approach to vanilla mixup and its variants, where it changes the probabilities of sampling mixing pairs instead of changing the way to mixing. We further conduct compatibility analysis of C-Mixup by integrating it to three representative mixup variants – PuzzleMix [32], CutMix [81], AutoMix [45]. We evaluate the performance on two datasets (RCF-MNIST, PovertyMap). The reported results in Table 4 indicate the compatibility and efficacy of C-Mixup in regression.

Table 4: Compatibility analysis. See Appendix F.1 for full results.

| Model | | RCF-MNIST | PovertyMap |
|---|---|---|---|
| | | RMSE ↓ | Worst $R$ ↑ |
| CutMix | | 0.194 | 0.46 |
| | +C-Mixup | **0.186** | **0.53** |
| PuzzleMix | | 0.159 | 0.47 |
| | +C-Mixup | **0.150** | **0.50** |
| AutoMix | | 0.152 | 0.49 |
| | +C-Mixup | **0.146** | **0.53** |

**II. Analysis of Distance Metrics.** Besides the theoretical analysis, here we empirically analyze the effectiveness of using label distance. Here, we use $d(a, b)$ to denote the distance between objects $a$ and $b$, e.g., $d(y_i, y_j)$ in our case. We consider four substitute distance metrics, including: (1) *feature distance*: $d(x_i, x_j)$; (2) *feature and label distance*: we concatenate the input feature $x$ and label $y$ to compute the distance, i.e., $d(x_i \oplus y_i, x_j \oplus y_j)$; (3) *representation distance*: since using feature distance may fail to capture high-level feature relations, we propose to use the model's hidden representations to measure the distance $d(h_i, h_j)$, detailed in Appendix F.2. (4) *representation and label distance*: $d(h_i \oplus y_i, h_j \oplus y_j)$.

Table 5: Performance of different distance metrics. $x$, $y$, $h$ represent input feature, label, and hidden representation, respectively. The full results are listed in Appendix F.2.

| Model | Ex.-Rate | Shape1D | DTI |
|---|---|---|---|
| | RMSE ↓ | MSE ↓ | Avg. $R$ ↑ |
| ERM/MAML | 0.0236 | 4.698 | 0.464 |
| mixup/MetaMix | 0.0239 | 4.275 | 0.465 |
| $d(x_i, x_j)$ | 0.0212 | 4.539 | 0.478 |
| $d(x_i \oplus y_i, x_j \oplus y_j)$ | 0.0212 | 4.395 | 0.484 |
| $d(h_i, h_j)$ | 0.0213 | 4.202 | 0.483 |
| $d(h_i \oplus y_i, h_j \oplus y_j)$ | 0.0208 | 4.176 | 0.487 |
| $d(y_i, y_j)$ **(C-Mixup)** | **0.0203** | **4.024** | **0.498** |

The results are reported in Table 5. We find that (1) feature distance $d(x_i, x_j)$ does benefit performance compared with ERM and vanilla mixup in all cases since it selects more reasonable example pairs for mixing; (2) representation distance $d(h_i, h_j)$ outperforms feature distance $d(x_i, x_j)$ since it captures high-level features; (3) though involving label in representation distance is likely to overwhelm the effect of labels, the better performance of $d(h_i \oplus y_i, h_j \oplus y_j)$ over $d(h_i, h_j)$ indicates the effectiveness of regression labels in measuring example distance, which is further verified by the superiority of C-Mixup. (4) Our discussion of computational efficiency in Appendix A.3 indicates that C-Mixup is much more computationally efficient than other distance measurements.

**III. How does the choice of bandwidth affects the performance?** Finally, we analyze the effects of the bandwidth $\sigma$ in Eqn. (6). The performance with respect to bandwidth of Exchange-Rate and DTI are visualized in Figure 5, respectively. We point out that if the bandwidth is too large, the results are close to vanilla mixup or Manifold Mixup, depending on which one is used. Otherwise, the results are close to ERM if the bandwidth is too small. According to the results, we see that the improvements of C-Mixup is somewhat stable over different bandwidths. Most importantly, C-Mixup yields a good model for

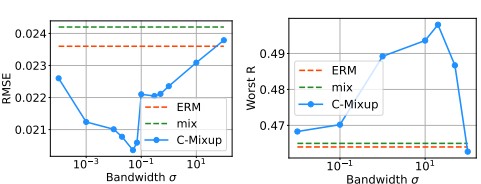

(a) : Exchange-Rate      (b) : DTI

Figure 5: Sensitivity analysis of bandwidth. mix represents the better mixing approach between mixup and Manifold Mixup.

a wide range of bandwidths, which reduces the efforts to tune the bandwidth for every specific dataset. Additionally, we provide empirical guidance about how to pick suitable bandwidth in Appendix F.3.1.

# 6 Related Work

**Data Augmentation and Mixup.** Various data augmentation strategies have been proposed to improve the generalization of deep neural networks, including directly augmenting images with manually designed strategies (e.g., whitening, cropping) [37], generating more examples with generative models [3, 7, 88, 85], and automatically finding augmentation strategies [10, 11, 43]. Mixup [82] and its variants [9, 12, 20, 24, 26, 32, 33, 44, 45, 52, 66–68, 81, 87] propose to improve generalization by linear interpolating input features of a pair of examples and their corresponding labels. Though mixup and its variants have demonstrated their power in classification [82], sequence labeling [84], and reinforcement learning [70], systematic analysis of mixup on different regression tasks is still underexplored. The recent MixRL [29] learns a policy network to select nearby example pairs for mixing, which requires substantial computational resources and is not suitable to high-dimensional real-world data. Unlike this method, C-Mixup instead adjusts the sampling probability of mixing pairs based on label similarity, which makes it much more efficient. Furthermore, C-Mixup, which focuses on how to select mixing examples, is a complementary method over mixup and its representative variants (see more discussion in Appendix A.4). Empirically, our experiments show the effectiveness and compatibility of C-Mixup on multiple regression tasks in Section 5.1.

**Task Generalization.** Our experiments extended C-Mixup to gradient-based meta-learning, aiming to improve task generalization. In the literature, there are two lines of related works. The first line of research directly imposes regularization on meta-learning algorithms [21, 30, 63, 79]. The second line of approaches introduces task augmentation to produce more tasks for meta-training, including imposing label noise [55], mixing support and query sets in the outer-loop optimization [49, 74], and directly interpolating tasks to densify the entire task distribution [77]. C-Mixup is complimentary to the latter mixup-based task augmentation methods. Furthermore, Section 5.2 indicates that C-Mixup empirically outperforms multiple representative prior approaches [79, 55].

**Out-of-Distribution Robustness.** Many recent methods aim to build machine learning models that are robust to distribution shift, including learning invariant representations with domain alignment [17, 41, 42, 46, 65, 72, 80, 86] or using explicit regularizers to finding a invariant predictors that performs well over all domains [1, 4, 23, 31, 36, 39, 75]. Recently, LISA [76] cancels out domain-associated correlations and learns invariant predictors by mixing examples either with the same label but different domains or with the same domain but different labels. While related, C-Mixup can be considered as a more general version of LISA that can be used for regression tasks. Besides, unlike LISA, C-Mixup do not use domain annotations, which are often expensive to obtain.

# 7 Conclusion

In this paper, we proposed C-Mixup, a simple yet effective variant of mixup that is well-suited to regression tasks in deep neural networks. Specifically, C-Mixup adjusts the sampling probability of mixing example pairs by assigning higher probability to pairs with closer label values. Both theoretical and empirical results demonstrate the promise of C-Mixup in improving in-distribution generalization, task generalization, and out-of-distribution robustness.

Theoretical future work may relax the assumptions in our theoretical analysis and extend the results to more complicated scenarios. We also plan to analyze more properties of C-Mixup theoretically, e.g., the relation between C-Mixup and manifold intrusion [22] in regression. Empirically, we plan to investigate how C-Mixup performs in more diverse application tasks such as semantic segmentation, natural language understanding, reinforcement learning.

# Acknowledgement

We thank Yoonho Lee, Pang Wei Koh, Zhen-Yu Zhang, and members of the IRIS lab for the many insightful discussions and helpful feedback. This research was funded in part by JPMorgan Chase & Co. Any views or opinions expressed herein are solely those of the authors listed, and may differ from the views and opinions expressed by JPMorgan Chase & Co. or its affiliates. This material is not a product of the Research Department of J.P. Morgan Securities LLC. This material should not be construed as an individual recommendation for any particular client and is not intended as a recommendation of particular securities, financial instruments or strategies for a particular client. This material does not constitute a solicitation or offer in any jurisdiction. The research was also supported by Apple, Intel and Juniper Networks. CF is a CIFAR fellow. The research of Linjun Zhang is partially supported by NSF DMS-2015378.

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
