# C-Mixup: Improving Generalization in Regression – Appendix

**Huaxiu Yao**[1][*], **Yiping Wang**[2][*], **Linjun Zhang**[3], **James Zou**[1], **Chelsea Finn**[1]

[1]Stanford University, [2]Zhejiang University, [3]Rutgers University

[1]{huaxiu,cbfinn}@cs.stanford.edu, jamesz@stanford.edu

[2]yipingwang6161@gmail.com, [3]linjun.zhang@rutgers.edu

## A    Additional Information for C-Mixup

### A.1    Illustration of How C-Mixup Improves Out-of-Distribution Robustness

In Figure 1, we use the ShapeNet1D to illustrate how C-Mixup improves out-of-distribution robustness. Here, we color the images to construct different domains. We train the model on red and blue domains and then generalize it to the green one. In Figure 1, we can see that C-Mixup can recognize more reasonable mixing pairs compared with vanilla mixup. Mixup with feature similarity fails to cancel out the domain information since it may be easier to mix unreasonable example pairs within the same domain. C-Mixup instead is naturally suitable to average out domain information by mixing examples with close labels.

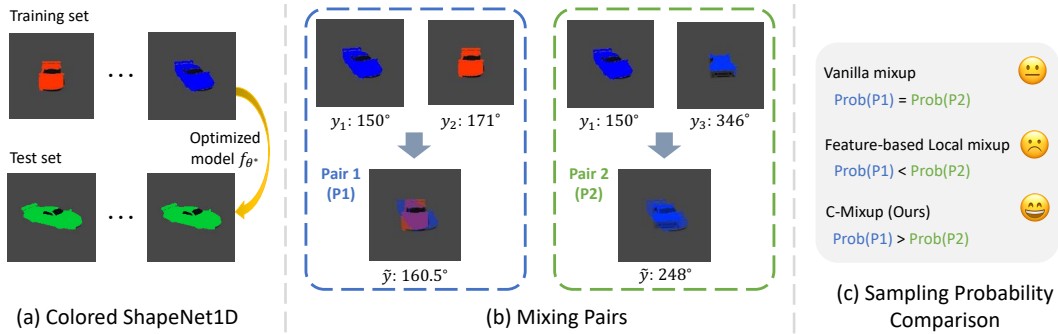

Figure 1: Illustration of C-Mixup for out-of-distribution robustness. Here, we color the ShapeNet1D and regard color as the domain information. $\lambda$ represents the interpolation ratio. (a) Colored ShapeNet1D pose prediction task, aiming to generalize the model trained on red and blue domains to the green domain; (b) Two mixing pairs with interpolated images and labels; (c) Illustration of a rough comparison of sampling probabilities between two mixing pairs in (b). Here, C-Mixup is capable of assigning higher sampling probability to more reasonable pairs and eliminate the effect of domain information.

### A.2    Algorithm of Meta-Training with C-Mixup

In this section, we summarize the algorithm of applying C-Mixup to MetaMix [17] in Alg. 1. Here, we adopt MetaMix with mixup version, which could be easily adapted to other mixup variants (e.g., CutMix, Manifold Mixup)

---

[*][*]Equal contribution. This work was done when Yiping Wang was remotely co-mentored by Huaxiu Yao and Linjun Zhang.

36th Conference on Neural Information Processing Systems (NeurIPS 2022).

---

**Algorithm 1** Meta-Training Process of MetaMix with C-Mixup

---

**Require:** Outer-loop learning rate $\eta$; Inner-loop learning rate $\xi$ (we change $\alpha$ in Eqn. (4) of the main paper to $\xi$ to avoid notation conflict); Shape parameter $\alpha$; Task distribution $p(\mathcal{T})$
1: Randomly initialize model parameters $\theta$
2: **while** not converge **do**
3:      Sample a batch of tasks $\{\mathcal{T}_i\}_{i=1}^{|M|}$ with the corresponding dataset $\mathcal{D}_m$
4:      **for** all $\mathcal{T}_m$ **do**
5:          Sample a support set $\mathcal{D}_m^s$ and a query set $\mathcal{D}_m^q$ from $\mathcal{D}_m$
6:          Calculate pairwise distance matrix $P$ between query set and support set via Eqn. (6).
7:          Calculate the task-specific parameter $\phi_m$ via the inner-loop gradient descent, i.e., $\phi_m = \theta - \xi \nabla_\theta \mathcal{L}(f_\theta; \mathcal{D}_m^s)$
8:          **for** each query example $(x_{m,i}^q, y_{m,i}^q)$ **do**
9:              Sample MetaMix parameter $\lambda \sim \mathrm{Beta}(\alpha, \alpha)$
10:         Sample support set example $(x_{m,j}^s, y_{m,j}^s)$ according to the probability $P(\cdot \mid (x_{m,i}^q, y_{m,i}^q))$
11:         Linearly interpolate $(x_{m,i}^q, y_{m,i}^q)$ and $(x_{m,j}^s, y_{m,j}^s)$ to get $(\tilde{x}_{m,i}^q, \tilde{y}_{m,i}^q)$
12:         Replace $(x_{m,i}^q, y_{m,i}^q)$ with $(\tilde{x}_{m,i}^q, \tilde{y}_{m,i}^q)$
13:      Use interpolated examples to update the model via $\theta \leftarrow \theta - \eta \frac{1}{|M|} \sum_{i=1}^{|M|} \mathcal{L}(f_{\phi_m}; \tilde{\mathcal{D}}_m^q)$

---

**Algorithm 2** Training with C-Mixup-batch

---

**Require:** Learning rates $\eta$; Shape parameter $\alpha$
**Require:** Training data $\mathcal{D} := \{(x_i, y_i)\}_{i=1}^N$
1: Randomly initialize model parameters $\theta$
2: **while** not converge **do**
3:      Sample two batches of examples $\mathcal{B}_1, \mathcal{B}_2 \sim \mathcal{D}$
4:      Calculate pairwise distance matrix $P$ between $\mathcal{B}_1$ and $\mathcal{B}_2$ via Eqn. (6)
5:      **for** each example $(x_i, y_i) \in \mathcal{B}_1$ **do**
6:          Sample example $(x_j, y_j)$ from $\mathcal{B}_2$ according to the probability $P(\cdot \mid (x_i, y_i))$
7:          Sample $\lambda$ from $\mathrm{Beta}(\alpha, \alpha)$
8:          Linearly interpolate $(x_i, y_i)$ and $(x_j, y_j)$ to get $(\tilde{x}, \tilde{y})$
9:      Use interpolated examples to update the model via Eqn. (3)

---

### A.3 Efficiency Discussion of C-Mixup

Assume the number of examples are $n$, the dimension of features and labels are $M_f$ and $M_l$, respectively. The time complexity of calculating the pairwise distance matrix $P$ with feature distance or label distance is $O(n^2 M_f)$ or $O(n^2 M_l)$, respectively. Generally, since $M_l \ll M_f$, using label distance (i.e., C-Mixup) substantially reduces the cost of calculating the pairwise distance matrix.

Furthermore, the calculation of pairwise distance matrix can be accelerated using parallelized operations, but it is still challenging if $n$ is sufficiently large, e.g., billions of examples. We thus propose an alternative solution that applies C-Mixup only to every example batch, which is named as **C-Mixup-batch**. In Alg. 2, we summarize the training process of C-Mixup-batch. We compare C-Mixup and C-Mixup-batch, and report the results of in-distribution generalization and out-of-distribution robustness in Table 1 and Table 2, respectively. Notice that we have used C-Mixup-batch for Echo, RCF-MNIST, ProvertyMap since calculating pair-wise distance metrics for these large datasets is time-consuming. Hence, we only report the results for other datasets. According to the results, we observe that C-Mixup-batch achieves comparable performance to C-Mixup. Nevertheless, the downside of C-Mixup-batch is that we must calculate the pairwise distance matrix for every batch. Accordingly, original C-Mixup is suitable to most datasets, while C-Mixup-batch is more appropriate to large datasets (e.g., Echo).

### A.4 Discussion between C-Mixup and Mixup

In this paper, we regard C-Mixup is an complementary approach to mixup and its most representative variants (e.g., Manifold Mixup [15], CutMix [19]). Here, we use vanilla mixup as an exemplar to

Table 1: Comparison between C-Mixup and C-Mixup-batch to in-distribution generalization. Since Echo is a large dataset and C-Mixup-batch is used by default, we only report the results of Airfoil, NO2, Exchange-Rate and Electricity here.

| | Dataset | Airfoil | NO2 | Exchange-Rate | Electricity |
|---|---|---|---|---|---|
| RMSE ↓ | C-Mixup-batch | $2.792 \pm 0.135$ | $0.510 \pm 0.007$ | $0.0205 \pm 0.0017$ | $0.0576 \pm 0.0002$ |
| | C-Mixup | $2.717 \pm 0.067$ | $0.509 \pm 0.006$ | $0.0203 \pm 0.0011$ | $0.0570 \pm 0.0006$ |
| MAPE ↓ | C-Mixup-batch | $1.616 \pm 0.053\%$ | $12.894 \pm 0.180\%$ | $2.064 \pm 0.218\%$ | $13.697 \pm 0.155\%$ |
| | C-Mixup | $1.610 \pm 0.085\%$ | $12.998 \pm 0.271\%$ | $2.041 \pm 0.134\%$ | $13.372 \pm 0.106\%$ |

Table 2: Comparison between C-Mixup and C-Mixup-batch to out-of-distribution robustness. Since we have applied C-Mixup-batch to PovertyMap, we report the results of Crime, SkillCraft, DTI.

| | Dataset | Crime (RMSE ↓) | SkillCraft (RMSE ↓) | DTI ($R$ ↑) |
|---|---|---|---|---|
| Avg | C-Mixup-batch | $0.125 \pm 0.001$ | $5.619 \pm 0.212$ | $0.490 \pm 0.005$ |
| | C-Mixup | $0.123 \pm 0.000$ | $5.201 \pm 0.059$ | $0.498 \pm 0.008$ |
| Worst | C-Mixup-batch | $0.152 \pm 0.007$ | $7.665 \pm 0.875$ | $0.453 \pm 0.006$ |
| | C-Mixup | $0.146 \pm 0.002$ | $7.362 \pm 0.244$ | $0.458 \pm 0.004$ |

show the difference. According to our description of mixup in Section 2 of the main paper, the entire mixup process includes three stages:

- Stage I: sample two instances $(x_i, y_i)$, $(x_j, y_j)$ from the training set.

- Stage II: sample the interpolation factor $\lambda$ from the Beta distribution Beta$(\alpha, \alpha)$.

- Stage III: mixing the sampled instances with interpolation factor $\lambda$ according to the following mixing formulation:

$$x_{mix} = \lambda x_i + (1 - \lambda)x_j, y_{mix} = \lambda y_i + (1 - \lambda)y_j, \lambda \sim \text{Beta}(\alpha, \alpha).$$

In the original mixup, the interpolation factor $\lambda$ sampled in the stage II controls how to mix these two instances. C-Mixup instead manipulates stage I and pairs with closer labels are more likely to be sampled.

In addition to the discussion of the complementarity of C-Mixup, the original mixup paper further shows that randomly interpolating examples from the same label performs worse than completely random mixing examples in classification. Compared to classification, randomly mixing examples in regression may be easier to generate semantically wrong labels. Intuitively, linearly mixing one-hot labels in classification is easy to generate semantically meaningful artificial labels, where the mixed label represents the probabilities of mixed examples to some extent. While in regression, the mixed labels may be semantically meaningless (e.g., pairs 2 and 3 in Figure **??**) and more significantly affect the performance. By mixing examples with closer labels, C-Mixup mitigates the influence of semantically wrong labels and improves the in-distribution and task generalization in regression. Additionally, C-Mixup further shows its superiority in improving out-of-distribution robustness in regression, which is not discussed in the original mixup paper.

# B  Detailed Proofs

In this section, we provide detailed proofs of Theorem 1, 2, 3. To avoid symbol conflict, we would like to point out that we use $h$ to denote the bandwidth of kernel in the nonparametric estimation step, which is different from the bandwidth $\sigma$ in Eqn. (6) of the main paper, which was used to measure the similarity in mixup. In Section B.4, we provide proofs for all used Lemmas.

## B.1 Proof of Theorem 1

We first state the kernel estimator.

$$\hat{g}(t; \theta) = \frac{\sum_{i=1}^{n} K(\theta^\top x_i, t) y_i}{\sum_{i=1}^{n} K(\theta^\top x_i, t)},$$

this kernel function can take, for example, the uniform kernel function $K(t_1, t_2) = 1\{|t_1 - t_2| < h\}$ or a Gaussian kernel function $K(t_1, t_2) = \exp(-|t_1 - t_2|^2/h^2)$ where $h$ is the bandwidth.

To make the proof easier to follow, we restate Theorem 1 below. Suppose $\theta \in \mathbb{R}^p$ is sparse with sparsity $s = o(\min\{p, \sigma_\xi^2\})$, $p = o(N)$ and $g$ is smooth with $c_0 < g' < c_1$, $c_2 < g'' < c_3$ for some universal constants $c_0, c_1, c_2, c_3 > 0$. There exists a distribution on $x$ with a kernel function, such that when the sample size $N$ is sufficiently large, with probability $1 - o(1)$,

$$\text{MSE}(\theta^*_{C-Mixup}) < \min(\text{MSE}(\theta^*_{feat}), \text{MSE}(\theta^*_{mixup})). \tag{1}$$

**Proof.** For identifiability, we consider the case where $\theta$ has $\ell_2$ norm of 1. Let us construct the distribution of $x$ as $z \sim \frac{1}{K} \sum_{k=1}^{K} N_p(\mu_k, \sigma_z^2 I_p)$ with $\mu_k = \frac{k}{\|\theta\|}\theta$ for a fixed positive integer $K$. We break out the entire proof into three steps.

**Step 1.** We first analyze the behavior of the three mixup methods. First, we have

$$\theta^\top z \sim \frac{1}{K} \sum_{k=1}^{K} N_p(\mu_k^\top \theta, \sigma_z^2 \|\theta\|^2),$$

and

$$\theta^\top \xi \sim \frac{1}{K} \sum_{k=1}^{K} N_p(0, \sigma_\xi^2 \|\theta\|^2).$$

We then have

$$|y_{k'} - y_k| = |g(\mu_k^\top \theta) - g(\mu_{k'}^\top \theta)| \pm \Theta((\sigma_\xi + \sigma_x)\|\theta\|) = |g(k) - g(k')| \pm \Theta((\sigma_\xi + \sigma_x)\|\theta\|)$$

$$\|x_{k'} - x_k\| = \|\mu_k - \mu_k'\| \pm \Theta((\sigma_\xi + \sigma_x) \cdot \sqrt{d}) = |k - k'| \pm \Theta((\sigma_\xi + \sigma_x) \cdot \sqrt{d})$$

As a result, if we take $(\sigma_\xi + \sigma_x)\|\theta\| = o(1)$, and $(\sigma_\xi + \sigma_x) \cdot \sqrt{d} \to \infty$, C-Mixup only interpolates examples within the same cluster, while mixup with feature similarity and vanilla mixup interpolate examples across different clusters.

**Step 2.** We then show that $\hat{\theta}$ obtained by all the three methods is consistent (the estimation error goes to 0 when the sample size goes to infinity). We first present a lemma showing that in expectation, the solution would recover $\theta$.

**Lemma 1.** *Suppose $x_i$'s are i.i.d. sampled from $N_p(\mu, I)$, then we have*

$$\mathbb{E}[y_i x_i] = (\mathbb{E}[g'(x_i^\top \theta)] + \mathbb{E}[g(x_i^\top \theta)]) \cdot \theta.$$

The proof of Lemma 1 is deferred to Section B.4.

This lemma implies that if $\tilde{x} = \lambda x_k + (1 - \lambda)x_k'$, $\tilde{y} = \lambda y_k + (1 - \lambda)y_k'$, we have

$$\mathbb{E}[\tilde{y}\tilde{x}] = c_{k,k'}\theta,$$

for some constant $c_{k,k'}$. Additionally, we have $\mathbb{E}[\tilde{x}\tilde{x}^\top] = cI + c'_{k,k'}\theta\theta^\top$. Therefore, $\mathbb{E}[\tilde{x}\tilde{x}^\top]^{-1}\mathbb{E}[\tilde{y}\tilde{x}] = \tilde{c}_{k,k'}\theta$ (via the Sherman–Morrison formula), for some constant $\tilde{c}_{k,k'}$.

Since we assume $g$ is $c_1$-Lipschitz for some universal constant $c_1$, which also implies $\tilde{c}_{k,k'} = O(1)$. We then analyze the convergence of $\hat{\theta}$. By definition, we have

$$\hat{\theta} = (\frac{1}{N} \sum_{i=1}^{N} \tilde{x}_i \tilde{x}_i^\top)^{-1}(\frac{1}{N} \sum_{i=1}^{N} \tilde{x}_i \tilde{y}_i).$$

Using Bernstein inequality, we have with probability at least $1 - p^{-2}$,

$$\|\frac{1}{N} \sum_{i=1}^{N} \tilde{x}_i \tilde{x}_i^\top - \mathbb{E}[\tilde{x}\tilde{x}^\top]\| = O(\sqrt{\frac{p}{N}}),$$

and

$$\|\frac{1}{N}\sum_{i=1}^{N}\tilde{x}_i\tilde{y}_i - \mathbb{E}[\tilde{x}\tilde{y}]\| = O(\sqrt{\frac{p}{N}}).$$

Then using Lemma 1, since $\lambda_{\min}(\mathbb{E}[\tilde{x}\tilde{x}^\top]) \gtrsim c_1$, when $n$ is sufficiently large, we then have

$$\|\hat{\theta} - \theta\|_2 \lesssim \sqrt{\frac{p}{N}} = o(1).$$

**Step 3.** We finally proceed to the nonparametric estimation step.

For C-Mixup, since we only interpolates the samples within the same Gaussian cluster, using the fact that $\sigma_\xi = o(1)$, $K$ being Lipschitz, and $\|\hat{\theta} - \theta\|_2 = o(1)$, we have that

$$\|\hat{g}(t;\hat{\theta}) - \frac{\sum_{i=1}^{N}K(z_i^\top\theta,t)y_i}{\sum_{i=1}^{N}K(z_i^\top\theta,t)}\| = \|\frac{\sum_{i=1}^{N}K(x_i^\top\hat{\theta},t)y_i}{\sum_{i=1}^{N}K(x_i^\top\hat{\theta},t)} - \frac{\sum_{i=1}^{N}K(z_i^\top\theta,t)y_i}{\sum_{i=1}^{N}K(z_i^\top\theta,t)}\| = o(1), \quad (2)$$

here the function norm of $h$ is defined as $\|h\| = \sqrt{\mathbb{E}[h^2(x)]}$.

Using the standard nonparametric regression results (e.g., see Tsybakov [14]), when the feature is observed without noise, the kernel estimator is consistent:

$$\|g(t) - \frac{\sum_{i=1}^{N}K(z_i^\top\theta,t)y_i}{\sum_{i=1}^{N}K(z_i^\top\theta,t)}\| = o(1). \tag{3}$$

Combining the two inequalities (2) and (3), we find that the $\hat{g}$ obtained by C-Mixup satisfies

$$\|\hat{g} - g\| = o(1).$$

For vanilla mixup and mixup with feature similarity, we have show that with a nontrivial positive probability, the samples are from two different clusters. Therefore using the assumption on $g'$ and $g''$, we have $\tilde{y} - y_i \geq c$ for some constant $c > 0$ with Jensen's inequality. As a result,

$$|\frac{\sum_{i=1}^{N}K(x_i^\top\hat{\theta},t)y_i}{\sum_{i=1}^{N}K(x_i^\top\hat{\theta},t)} - \frac{\sum_{i=1}^{N}K(x_i^\top\hat{\theta},t)\tilde{y}_i}{\sum_{i=1}^{N}K(x_i^\top\hat{\theta},t)}| \geq c.$$

Combining with the two inequalities (2) and (3), we have the $\hat{g}$ obtained by vanilla mixup or mixup with feature similarity satisfies

$$\|\hat{g} - g\| > c.$$

Since $\text{MSE}(\theta) \asymp \|\hat{g}(\cdot;\theta) - g(\cdot)\|$, we then have

$$\text{MSE}(\theta^*_{C-Mixup}) < \min(\text{MSE}(\theta^*_{feat}), \text{MSE}(\theta^*_{mixup})).$$

## B.2 Proof of Theorem 2

We first restate Theorem 2. Let $N = \sum_{m=1}^{M} N_m$ and $N_m$ is the number of examples of $\mathcal{T}_m$. Suppose $\theta_k$ is sparse with sparsity $s = o(\min\{d, \sigma_\xi^2\})$, $p = o(N)$ and $g_m$'s are smooth with $0 < g'_m < c_1$, $c_2 < g''_m < c_3$ for some universal constants $c_1, c_2, c_3 > 0$ and $m \in [M] \cup \{t\}$. There exists a distribution on $x$ with a kernel function, such that when the sample size $N$ is sufficiently large, with probability $1 - o(1)$,

$$\text{MSE}_{\text{Target}}(\theta^*_{Meta-C-Mixup}) < \min(\text{MSE}_{\text{Target}}(\theta^*_{Meta-feat}), \text{MSE}_{\text{Target}}(\theta^*_{MetaMix})). \tag{4}$$

Again, we consider the distribution of $x$ to be $z \sim \frac{1}{K}\sum_{k=1}^{K}N_p(\mu_k, \sigma_z^2 I_p)$ with $\mu_k = \frac{k}{\|\theta\|}\theta$ for a fixed positive integer $K$. The proof of Theorem 2 largely follows Theorem 1, with the only difference in the step 2. We prove the step 2 for Theorem 2 in the following.

By Lemma 1, for augmented data in the $m$-th task, $\tilde{x}^{(m)} = \lambda x_k^{(m)} + (1-\lambda)x_k^{(m)'}$, $\tilde{y}^{(m)} = \lambda y_k^{(m)} + (1-\lambda)y_k^{(m)'}$, we have

$$\mathbb{E}[\tilde{y}^{(m)}\tilde{x}^{(m)}] = c_{k,k'}^{(m)}\theta,$$

for some constant $c_{k,k'}^{(m)}$.

Additionally, we have $\mathbb{E}[\tilde{x}^{(m)}\tilde{x}^{(m)\top}] = c^{(m)}I + c_{k,k'}^{(m)'}\theta\theta^\top$. Therefore, $(\sum_{m=1}^T \mathbb{E}[\tilde{x}^{(m)}\tilde{x}^{(m)\top}])^{-1}(\sum_{m=1}^T \mathbb{E}[\tilde{y}^{(m)}\tilde{x}^{(m)}]) = \tilde{c}_{k,k'}\theta$, for some constant $\tilde{c}_{k,k'}$.

We then analyze the convergence of $\hat{\theta}$. By definition, we have

$$\hat{\theta} = (\frac{1}{N}\sum_{m=1}^T \frac{1}{n_m}\sum_{i=1}^{n_t} \tilde{x}_i^{(m)}\tilde{x}_i^{(m)\top})^{-1}(\frac{1}{N}\sum_{m=1}^T \frac{1}{n_m}\sum_{i=1}^{n_t} \tilde{x}_i^{(m)}\tilde{y}_i^{(m)}).$$

Using Bernstein inequality, we have with probability at least $1 - p^{-2}$,

$$\|\frac{1}{N}\sum_{m=1}^T \frac{1}{n_m}\sum_{i=1}^{n_m} x_i^{(m)}x_i^{(m)\top} - \frac{1}{N}\sum_{m=1}^T \mathbb{E}[\tilde{x}^{(m)}\tilde{x}^{(m)\top}]\| = O(\sqrt{\frac{p}{N}}),$$

and

$$\|\frac{1}{T}\sum_{m=1}^T \frac{1}{n_m}\sum_{i=1}^{n_m} x_i^{(m)}y_i^{(m)} - \mathbb{E}[\frac{1}{T}\sum_{t=1}^T \mathbb{E}[x^{(m)}y^{(m)}]]\| = O(\sqrt{\frac{p}{N}}).$$

Then using Lemma 1, when $N$ is sufficiently large, we then have

$$\|\hat{\theta} - \theta\|_2 \lesssim \sqrt{\frac{p}{N}} = o(1).$$

## B.3 Proof of Theorem 3

Similarly we restate Theorem 3.

Supposed for some $\max(\exp(-n^{1-o(1)}), \exp(-\frac{p_1^2}{2n})) < \delta \ll 1$, we have variance constraints: $\sigma_a = c_1\sigma_x$, $\sigma_x \geq c_2 \max(\frac{n^{5/2}}{\|\theta\|\delta}\sigma_\epsilon, \frac{\sqrt{p_2}\|\theta\|}{\sqrt{n}p_1})$ and $\sigma_\epsilon^2 \leq \frac{c_3}{pn^{3/2}}$. Then for any penalty k satisfies $c_4\sqrt{\frac{p_2}{p_1}}n^{1/4+o(1)} < k < c_5 \min(\frac{\sigma_x}{\|\theta\|}\sqrt{p_1 n^{1-o(1)}}, n)$ and bandwidth h satisfies $0 < h \leq c_6\frac{l}{\sqrt{\log(n^2/p_1)}}$ in C-Mixup, when $n$ is sufficiently large, with probability at least $1 - o(1)$, we have

$$\text{MSE}(\theta_{C-Mixup}^*) < \min(\text{MSE}(\theta_{feat}^*), \text{MSE}(\theta_{mixup}^*)), \tag{5}$$

where $c_1 \geq 1$, $c_2, c_3, c_4, c_5, c_6 > 0$ are universal constants, $l = \min_{i\neq j}|y_i - y_j'|$ and $p_1 \ll n < p_1^2$.

Let $p = p_1 + p_2$, and $\theta_1$ represents the subvectors that contain the first $p_1$ coordinates of $\theta$. Furthermore, let $\hat{X} \in \mathbb{R}^{n\times p}$ be an arbitrary noise-less data matrix and $\hat{\lambda}_0 \geq \hat{\lambda}_1 \geq ... \geq \hat{\lambda}_p$ be the singular values of $\hat{X}$. Similarly, let $E \in \mathbb{R}^{n\times p}$ be the noise matrix which contains $iid$. sub-Gaussian entries with variance proxy $\sigma_\epsilon^2$, and $\lambda_0 \geq \lambda_1 \geq ... \geq \lambda_p$ be the singular values of input matrix $X = \hat{X} + E$.

Firstly we show that the noise matrix only makes a small difference between the singular values of noise-less data matrix $\hat{X}$ and that of input matrix $X$.

**Lemma 2.** *If $e^{-n} < \delta_1 \ll 1$ and $\sigma_\epsilon^2 \leq \frac{c}{pn^{3/2}}$, then with probability at least $1 - \delta_1$ we have:*

$$|\hat{\lambda}_u - \lambda_u| \leq 16cn^{-1/2}, \tag{6}$$

*for every u satisfies $1 \leq u \leq p$.*

Before analyzing the effectiveness of C-Mixup, we first present the following lemma to analyze the C-Mixup with truncated label distance measurements. Specifically, we only apply C-Mixup to examples within a label distance threshold.

**Lemma 3.** *Assume $\exp(-n^{1-o(1)}) < \delta_1 \ll 1$, $\sigma_x \geq c_2\frac{n^{5/2}}{\|\theta\|\delta}\sigma_\epsilon$ for some $c_2$ that satisfies $c_{gap} := \frac{c_2\sqrt{\pi}}{4\sqrt{2+|\theta_1|^2}} > 1$. Here $c_{gap}$ is the ratio of $\min_{i\neq j}|y_i - y_j'|$ to $\max_i|y_i - y_i'|$. Then if we use C-Mixup, there exists some thresholds such that with probability at least $1 - \delta_1$, the training data $(x_i, y_i)$ will only be mixed with $(x_i', y_i')$. Here, we point out that $x_i$ and $x_i'$ are defined in Line 206-207 in the main paper.*

Then we consider replacing the truncated kernel with the gaussian kernel, which applies C-Mixup to all the examples with a smoother probability distribution. And we claim that there exist some bandwidths such that the data pairs with almost identical invariant features and opposite domain-changeable features will be mixed up together with high probability.

**Lemma 4.** *Assume $\exp(-\frac{p_1^2}{2n}) < \delta_1 \ll 1$ and for $l = \min_{i \neq j} |y_i - y_j'|$, we have $0 < h \leq c_6 \frac{l}{\sqrt{\log(n^2/p_1)}}$ for some $c_6 : 0 < c_6 \leq c_{gap}^{-1} \sqrt{(c_{gap}^2 - 1)/4}$, where $c_{gap}$ follows the definition in Lemma 3. We define the mixed input as $\hat{X}$, and let $S_i$ be a random variable that denotes if $(x_i, y_i)$ is mixed with $(x_i', y_i')$. Then with probability at least $1 - \delta_1$, we have:*

$$n - \frac{p_1}{2} \leq \sum_{i=1}^{n} S_i \leq n.$$

Next, we find that the input matrix $X$ corresponding to C-Mixup will have just $p_1$, rather than $p$, singular values that are much bigger than zero.

**Lemma 5.** *Assume the conditions of Lemma 2 and Lemma 4 still hold and the mixup ratio $\lambda = 0.5$. If noise-less data matrix $\hat{X}$ is obtained by C-Mixup, then the singular values of input matrix $X$ satisfy:*

$$(1 - b(n))^2 n \leq \lambda_i \leq (1 + b(n))^2 n, \qquad 1 \leq i \leq p_1,$$
$$|\lambda_i| \leq 16 \frac{c}{\sqrt{n}}, \qquad p_1 \leq i \leq p. \tag{7}$$

Finally, we can complete the proof of Theorem B.3 according to the lemmas above.

**Proof of Theorem 3.** Denote the input matrix as $X \in \mathbb{R}^{n \times p}$ and its singular values as $\lambda_0 \geq \lambda_1 \geq \ldots \geq \lambda_p$. Then, for ridge estimator with penalty $k$, we have Hoerl and Kennard [6] :

$$\mathbb{E}[|\theta^*(k) - \theta|^2] = \mathbb{E}[|(X^T X + kI)^{-1} X^T Y - \theta|^2]$$
$$= \sigma_x^2 \sum_{i=1}^{p} \frac{\lambda_i}{(\lambda_i + k)^2} + k^2 \theta^T (X^T X + kI)^{-2} \theta$$
$$= \sigma_x^2 \sum_{i=1}^{p_1} \frac{\lambda_i}{(\lambda_i + k)^2} + \sigma_x^2 \sum_{i=p_1+1}^{p_1+p_2} \frac{\lambda_i}{(\lambda_i + k)^2} + k^2 \theta^T (X^T X + kI)^{-2} \theta$$
$$= \gamma_1(k) + \gamma_2(k) + \gamma_3(k)$$

For the first term, we have

$$\gamma_1^{C-Mixup}(k) = \sigma_x^2 \sum_{i=1}^{p_1} \frac{\lambda_i}{(\lambda_i + k)^2}$$
$$\leq \sigma_x^2 \sum_{i=1}^{p_1} \frac{1}{(\min_{1 \leq i \leq p_1} \lambda_i)}$$
$$\leq \frac{\sigma_x^2 p_1}{(1 - b(n))^2 n}, \qquad \text{(Lemma 5)}$$

and

$$\gamma_1^{mixup}(k) \wedge \gamma_1^{feat}(k) \wedge \gamma_1^{C-Mixup}(k) \geq \sigma_x^2 \sum_{i=1}^{p_1} \frac{1}{4k} \geq \frac{\sigma_x^2 p_1}{4c_5 n}. \qquad (k < c_5 n, c_5 \geq \frac{1}{4})$$

To bound the last term, we perform orthogonal decompose on $X^T X$, i.e., $X^T X = P^T \Lambda P$. $P$ is orthogonal transformation and we denote $\alpha = P\theta$. Since the last $p_2$ coordinates of $\theta$ are 0, with probability at least $1 - \delta_1$, for C-Mixup we have:

$$\gamma_3^{C-Mixup}(k) = k^2 \theta^T (X^T X + kI)^{-2} \theta$$

$$= k^2 \sum_{i=1}^{p} \frac{\alpha_i^2}{(\lambda_i + k)^2}$$

$$= \sum_{i=1}^{p_1} \frac{\alpha_i^2}{(\lambda_i/k + 1)^2}$$

$$\leq \frac{k^2 |\theta|^2}{(\min_{1 \leq i \leq p_1} \lambda_i)^2} \qquad (\sum_{i=1}^{p_1} \alpha_i^2 = |\theta|^2)$$

$$\leq \frac{c_5^2}{(1 - b(n))^4 n^{o(1)}} \frac{\sigma_x^2 p_1}{n} \qquad (k < c_5 \frac{\sigma_x}{\|\theta\|} \sqrt{p_1 n^{1-o(1)}}, \text{ Lemma 5}).$$

For the second term, we bound C-Mixup as

$$\gamma_2^{C-Mixup}(k) = \sigma_x^2 \sum_{i=p_1+1}^{p_1+p_2} \frac{\lambda_i}{(\lambda_i + k)^2}$$

$$\leq \sigma_x^2 \sum_{i=p_1+1}^{p_1+p_2} \frac{\lambda_i}{k^2} \tag{8}$$

$$\leq \frac{16c}{c_4^2 n^{o(1)}} \frac{\sigma_x^2 p_1}{n}. \qquad (k > c_4 \sqrt{\frac{p_2}{p_1}} n^{1/4+o(1)}, \text{ Lemma 5})$$

For mixup and mixup with feature similarity, we bound the second term as:

$$\gamma_2^{feat}(k) \wedge \gamma_2^{mixup}(k) \geq \sigma_x^2 \sum_{i=p_1+1}^{p_1+p_2} \frac{1}{4k} \geq \frac{\sigma_x^2 p_2}{4c_5 n}. \qquad (k < c_5 n)$$

Thus there exists some constants $c, c_4, c_5$ (for example, $c_4 = 4\sqrt{c}$ and $c_5 = 1/4$), such that when n is sufficiently large, with probability at least $1 - o(1)$:

$$\mathbb{E}|\theta_{C-Mixup}^*(k) - \theta|^2 < \min(\mathbb{E}|\theta_{feat}^*(k) - \theta|^2, \mathbb{E}|\theta_{mixup}^*(k) - \theta|^2).$$

which can reduce to the results immediately.

### B.4 Proofs of Lemmas

**Proof of Lemma 1.** In order to prove Lemma 1, let us invoke the First-order Stein's Identity [13].

**Lemma 6.** *Let $X \in \mathbb{R}^d$ be a real-valued random vector with density p. Assume that p: $\mathbb{R}^d \to R$ is differentiable. In addition, let $g : \mathbb{R}^d \to \mathbb{R}$ be a continuous function such that $\mathbb{E}[\nabla g(X)]$ exists. Then it holds that*

$$\mathbb{E}[g(X) \cdot S(X)] = \mathbb{E}[\nabla g(X)],$$

*where $S(X) = -\nabla p(x)/p(x)$ is the score function of p.*

Now, let us plug in the density of $N_{d_1}(0, \Sigma_X)$, $p(x) = ce^{x^\top \Sigma_X^{-1} x/2}$ for some constant $c$. We then have $\nabla p(x) = ce^{x^\top \Sigma_X^{-1} x/2} \cdot \Sigma_X^{-1} x$ and $\nabla p(x)/p(x) = \Sigma_X^{-1} x$.

As a result, we have

$$\mathbb{E}[p^*(x)\Sigma_X^{-1} x] = \mathbb{E}[\nabla p^*(x)],$$

implying

$$\mathbb{E}[p^*(x)x] = \Sigma_X \mathbb{E}[\nabla p^*(x)].$$

Then recall that $p^*(x) = g(\theta^\top x)$, so we have $\nabla p^*(x) = g'(\theta^\top x)\theta$. Combining all the piece, we obtain

$$\mathbb{E}[p^*(x)x] = \Sigma_X \mathbb{E}[g'(\theta^\top x)]\theta.$$

Now plugging in $x \sim N(\theta, I)$, we have

$$\mathbb{E}[y_i x_i] = (\mathbb{E}[g'(x_i^\top \theta)] + \mathbb{E}[g(x_i^\top \theta)]) \cdot \theta.$$

**Proof of Lemma 2.** Denote the entry of $E$ as $\epsilon_{ij}$, $\epsilon_{ij} \sim subG(\sigma_\epsilon^2)$, i.e. sub-gaussian distribution with variance proxy $\sigma_\epsilon^2$. From Rigollet and Hütter [12], we find $\epsilon_{ij}^2 \sim subE(16\sigma_\epsilon^2)$, i.e. sub-exponential distribution with variance proxy $16\sigma_\epsilon^2$. Thus we choose $\delta_1$ from the Bernstein's inequality:

$$\mathbb{P}(\frac{1}{np} \sum_{ij} \epsilon_{i,j}^2 > t) \leq \exp[-\frac{np}{2} \min(\frac{t^2}{(16\sigma_\epsilon^2)^2}, \frac{t}{16\sigma_\epsilon^2})] = \delta_1 \tag{9}$$

Since $p_1, p_2 \geq 1$ we get $\delta_1 > e^{-n} \geq e^{-np/2}$, then:

$$t = 16\sigma_\epsilon^2 \max(\sqrt{\frac{2}{np} \log \frac{1}{\delta_1}}, \frac{2}{np} \log \frac{1}{\delta_1})$$
$$= 16\sigma_\epsilon^2 \sqrt{\frac{2}{np} \log \frac{1}{\delta_1}}$$
$$\leq 16\sigma_\epsilon^2$$

Thus, based on $\sigma_\epsilon^2 \leq \frac{c_3}{pn^{3/2}}$, with probability at least $1 - \delta_1$, we have:

$$\|E\|_F = \sum_{i,j} \epsilon_{ij}^2 \leq npt \leq 16\frac{c}{\sqrt{n}},$$

where $\| \cdot \|_F$ represents Frobenius norm. Then, with Hoffman-Weilandt's inequality [12] we prove that:

$$\max_u |\hat{\lambda}_u - \lambda_u| \leq \|E\|_F$$

**Proof of Lemma 3.** Since $z_i \sim \mathcal{N}_{p_1}(0, \sigma_x^2 I_{p_1})$, we have $(z_i - z_j)^T \theta_1 \sim \mathcal{N}(0, 2\sigma_x^2 |\theta_1|^2)$ for every $i \neq j$. Then for $t$ satisfies $0 < t \ll 1$, we have:

$$p_{ij} := \mathbb{P}(|(z_i - z_j)^T \theta_1| < t)$$
$$\leq 2\Phi(\frac{t}{\sqrt{2}\sigma_x |\theta|}) - 1 \tag{10}$$
$$= \frac{t}{\sqrt{\pi}\sigma_x |\theta|} + o(t^2),$$

where $\Phi(u) = \frac{1}{\sqrt{2\pi}} \int_{-\infty}^u e^{-q^2/2} dq$ is the distribution function of standard normal distribution. Thus if $t = \frac{\sqrt{\pi}\sigma_x |\theta|}{n(n-1)} \delta_1$, we have:

$$\mathbb{P}(\min_{i \neq j} |(z_i - z_j')^T \theta_1| \geq t) = 1 - \mathbb{P}(\min_{i \neq j} |(z_i - z_j')^T \theta_1| \leq t)$$
$$= 1 - \mathbb{P}(\bigcup_{i \neq j} \{|(z_i - z_j')^T \theta_1| \leq t\})$$
$$\geq 1 - \sum_{j \neq i} \sum_{i=1}^n p_{ij} \tag{11}$$
$$\geq 1 - \frac{n(n-1)}{2}(\frac{2t}{\sqrt{\pi}\sigma_x |\theta|}) \qquad \text{(Eqn. (10))}$$
$$= 1 - \delta_1.$$

On the other hand, $\max_i |y_i - y_i'| = \max_i |\theta_1^T \epsilon_i' + \epsilon_i - \epsilon_{i'}|$, and for every $i$, $(\theta_1^T \epsilon_i' + \epsilon_i - \epsilon_{i'}) \sim subG((2 + |\theta_1|^2)\sigma_\epsilon^2)$, then by maximum inequality [12], when $n$ is sufficiently large, with probability

at least $1 - \delta_1$:

$$\max_i |y_i - y_i'| \leq \sqrt{2(2 + |\theta_1|^2)\log(n/\delta_1)} \cdot \sigma_\epsilon$$

$$\leq 2\sigma_\epsilon \sqrt{(2 + |\theta_1|^2)n} \qquad (e^{-n} < \delta_1 \ll 1)$$

$$\leq \frac{2\sqrt{2 + |\theta_1|^2}}{c_2\sqrt{\pi}} t \qquad (\sigma_x \geq c_2 \frac{n^{5/2}}{\|\theta\|\delta}\sigma_\epsilon)$$

$$\leq \frac{2\sqrt{2 + |\theta_1|^2}}{c_2\sqrt{\pi}} \min_{i \neq j}|(z_i - z_j')^T\theta_1| \qquad \text{(Eqn. (11))} \qquad (12)$$

$$\leq \frac{2\sqrt{2 + |\theta_1|^2}}{c_2\sqrt{\pi}} (\min_{i \neq j}|y_i - y_j'| + \sqrt{2\log n}\sigma_\epsilon)$$

$$\leq \frac{4\sqrt{2 + |\theta_1|^2}}{c_2\sqrt{\pi}} \min_{i \neq j}|y_i - y_j'| \qquad (t \geq c_2\sqrt{\pi n}\sigma_\epsilon \geq 2\sqrt{2\log n}\sigma_\epsilon)$$

Choose $c_2$ such that $c_{gap} := \frac{c_2\sqrt{\pi}}{4\sqrt{2 + |\theta_1|^2}} > 1$ and denote $l = \min_{i \neq j}|y_i - y_j'|$, then we finish the proof with feasible threshold range $\frac{l}{c_{gap}} < h < l$.

**Proof of Lemma 4.** We define $Q_i = 1 - S_i$. From C-Mixup, we obtain:

$$P((x_j', y_j')|(x_i, y_i)) \propto \exp\left(-\frac{(y_i - y_j')^2}{2h^2}\right)$$

Then, by $\max_i |y_i - y_i'| \leq \frac{l}{c_{gap}}$, we have

$$\mathbb{E}(S) \geq \frac{1}{K}\exp(-\frac{l^2}{2h^2 c_{gap}^2})$$

$$\mathbb{E}(Q) \leq \frac{1}{K}(n-1)\exp(-\frac{l^2}{2h^2})$$

where $K = \sum_{j=1}^n (\exp(-(y_j' - y_i)^2/(2h^2)))$ is used for normalization. The upper bound of bandwidth $h$ is:

$$h < c_6\frac{l}{\sqrt{\log(n^2/p_1)}} < l \cdot \sqrt{\frac{c_{gap}^2 - 1}{2c_{gap}^2}}\log^{-\frac{1}{2}}(\frac{(n-1)(n-p_1/4)}{p_1/4}).$$

Since $S_i + Q_i = 1$, we obtain

$$\mathbb{E}(S) \geq \frac{n - p_1/4}{n}, \qquad \mathbb{E}(Q) \leq \frac{p_1/4}{n}.$$

Finally, since $S_i, Q_i \in [0, 1]$, we apply Hoeffding's inequality and obtain:

$$\mathbb{P}(\frac{1}{n}\sum_{i=1}^n S_i - \mathbb{E}(S) < -t) \leq \exp(-2nt^2) = \delta_1$$

Then with probability at least $1 - \delta_1$, we have:

$$\sum_{i=1}^n S_i \geq n(\mathbb{E}(S) - \sqrt{\frac{1}{2n}\log(\frac{1}{\delta_1})})$$

$$\geq n - \frac{p_1}{2}$$

**Proof of Lemma 5** For C-Mixup, according to Lemma 4, the corresponding noise-less matrix $\hat{X}$ is close to $(Z, O)$ when $n$ is sufficient large. Here, $Z \in \mathbb{R}^{n \times p_1}$ has rows $z_i$ and $O \in \mathbb{R}^{n \times p_2}$ is a matrix with at most $rank(\max(\frac{1}{2}p_1, p_2))$. We now simplify $O$ to be a zero matrix. In fact, Eqn. 8 in the

following proof just scale at most $3/2$ when $O$ is $rank(\max(\frac{1}{2}p_1, p_2))$, which does not affect the final results. From Theorem 4.6.1 in Vershynin [16], we find that there exists some positive absolute constants, such that with probability at least $1 - \delta_1$,

$$\sqrt{n} - C(\sqrt{p_1} + \sqrt{\log(2/\delta_1)}) \leq \hat{\lambda}_{p_1} \leq \hat{\lambda}_1 \leq \sqrt{n} + C(\sqrt{p_1} + \sqrt{\log(2/\delta_1)})$$

And $\hat{\lambda}_i = 0$ for $p_1 < i \leq p$. From Eqn.(6) and $\log(1/\delta_1) < n^{1-o(1)}$, we get Eqn. (7).

## C    Additional Experiments of In-Distribution Generalization

### C.1    Detailed Dataset Description

In this section, we provide detailed descriptions of datasets used in the experiments of in-distribution generalization.

**Airfoil Self-Noise [3]** contains aerodynamic and acoustic test results for different sizes NACA 0012 airfoils at various wind tunnel speeds and angles of attack. Specifically, each input have 5 features, including frequency, angle of attack, chord length, free-stream velocity, and suction side displacement thickness. The label is one-dimensional scaled sound pressure level. Min-max normalization is used to normalize input features. Follow [8], the number of examples in training, validation, and test sets are 1003, 300, and 200, respectively.

**NO2.** The NO2 emission dataset [1] originated from the study where air pollution at a road is related to traffic volume and meteorological variables. Each input contains 7 features, including logarithm of the number of cars per hour, temperature 2 meter above ground, wind speed, temperature difference between 25 and 2 meters above ground, wind direction, hour of day and day number from October 1st. 2001. The hourly values of the logarithm of the concentration of NO2, which was measured at Alnabru in Oslo between October 2001 and August 2003, are used as the response variable, i.e., the label. Follow [8], the number of training, validation and test sets are 200, 200 and 100, respectively.

**Exchange-Rate** is a time-series dataset that contains the collection of the daily exchange rates of 8 countries, including Australia, British, Canada, Switzerland, China, Japan, New Zealand and Singapore ranging from 1990 to 2016. The length of the entire time series is 7,588, and they adopt daily sample frequency. The slide window size is 168 days. The input dimension is $168 \times 8$ and the label dimension is $1 \times 8$ data. The dataset has been split into training ($60\%$), validation set ($20\%$) and test set ($20\%$) in chronological order as used in Lai et al. [10].

**Electricity** [4] is also a time-series dataset collected from 321 clients, which covers the electricity consumption in kWh every 15 minutes from 2012 to 2014. The length of the entire time-series is 26,304 and we use the hourly sample rate. Similar to Exchange-Rate data, the window size is set to 168, thus the input dimension is $168 \times 321$ the corresponding label dimension is $1 \times 321$. The dataset is also split as Lai et al. [10].

**Echocardiogram Videos (Echo)** [11] includes 10,030 apical-4-chamber labeled echocardiogram videos from different aspects and human expert annotations to study cardiac motion and chamber sizes. These videos were collected from individuals who underwent imaging at Stanford University Hospital between 2016 and 2018. To identify the area of the left ventricle, we first preprocess the videos with frame-by-frame semantic segmentation. This method outputs video clips that contain 32 frames of $112 \times 112$ RGB images, which are be used to predict ejection fraction. The entire dataset are split into training, validation and test sets with size 7,460, 1,288, and 1,276, respectively.

### C.2    Hyperparameters

We list the hyperparameters for every dataset in Table 3. Here, as we mentioned in Line 235-236 in the main paper, we apply k-Mixup, Local Mixup, MixRL, and C-Mixup to both mixup and Manifold Mixup, and report the best-performing one. Thus, we also treat the mixup type as another hyperparameter. All hyperparameters are selected by cross-validation. In addition, in Section F.3.1 of Appendix, we provide some guidance about how to tune and pick bandwidth $\sigma$. The guidance is also suitable to tasks beyond in-distribution generalization, i.e., task generalization and out-of-distribution robustness.

Table 3: Hyperparameter settings for the experiments of in-distribution generalization. Here, FCN3 means 3-layer fully connected network and ManiMix means Manifold Mixup.

| Dataset | Airfoil | NO2 | Exchange-Rate | Electricity | Echo |
|---|---|---|---|---|---|
| Learning rate | 1e-2 | 1e-2 | 1e-3 | 1e-3 | 1e-4 |
| Weight decay | 0 | 0 | 0 | 0 | 1e-4 |
| Scheduler | n/a | n/a | n/a | n/a | StepLR |
| Batch size | 16 | 32 | 128 | 128 | 10 |
| Type of mixup | ManiMix | mixup | ManiMix | mixup | mixup |
| Architecture | FCN3 | FCN3 | LST-Attn | LST-Attn | EchoNet-Dynamic |
| Horizon | n/a | n/a | 12 | 24 | n/a |
| Optimizer | Adam | Adam | Adam | Adam | Adam |
| Maximum Epoch | 100 | 100 | 100 | 100 | 20 |
| Bandwidth $\sigma$ | 1.75 | 1.2 | 5e-2 | 0.5 | 50.0 |
| $\alpha$ in Beta Dist. | 0.5 | 2.0 | 1.5 | 2.0 | 2.0 |

## C.3 Overfitting

In Figure 2, we visualize additional overfitting analysis on Electricity and the results corroborate our findings in the main paper, where C-Mixup reduces the generalization gap and achieves better test performance.

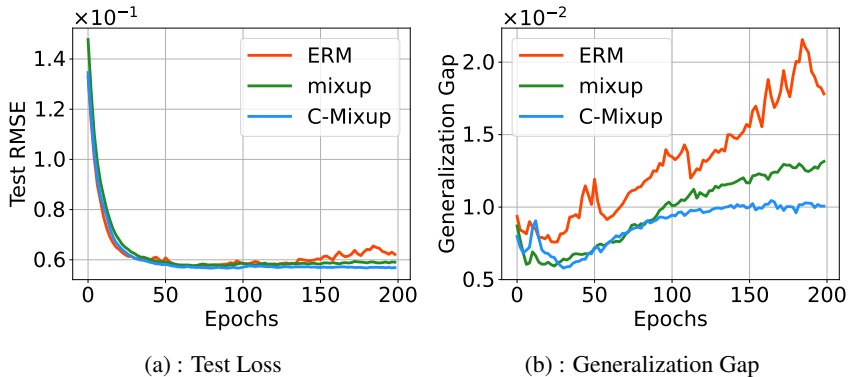

(a) : Test Loss    (b) : Generalization Gap

Figure 2: Additional overfitting analysis of electricity

## C.4 Full Results

In Table 4, we report the full results of in-distribution generalization.

Table 4: Full results of in-distribution generalization. We compute the mean and standard deviation for results of three seeds.

| | | Airfoil | NO2 | Exchange-Rate | Electricity | Echo |
|---|---|---|---|---|---|---|
| RMSE | ERM | $2.901 \pm 0.067$ | $0.537 \pm 0.005$ | $0.0236 \pm 0.0031$ | $0.0581 \pm 0.0011$ | $5.402 \pm 0.024$ |
| | mixup | $3.730 \pm 0.190$ | $0.528 \pm 0.005$ | $0.0239 \pm 0.0027$ | $0.0585 \pm 0.0004$ | $5.393 \pm 0.040$ |
| | Mani mixup | $3.063 \pm 0.113$ | $0.522 \pm 0.008$ | $0.0242 \pm 0.0043$ | $0.0583 \pm 0.0004$ | $5.482 \pm 0.066$ |
| | k-Mixup | $2.938 \pm 0.150$ | $0.519 \pm 0.005$ | $0.0236 \pm 0.0029$ | $0.0575 \pm 0.0002$ | $5.518 \pm 0.034$ |
| | Local Mixup | $3.703 \pm 0.151$ | $0.517 \pm 0.004$ | $0.0236 \pm 0.0024$ | $0.0582 \pm 0.0004$ | $5.652 \pm 0.043$ |
| | MixRL | $3.614 \pm 0.293$ | $0.527 \pm 0.003$ | $0.0238 \pm 0.0037$ | $0.0585 \pm 0.0006$ | $5.618 \pm 0.071$ |
| | **C-Mixup (Ours)** | $\mathbf{2.717 \pm 0.067}$ | $\mathbf{0.509 \pm 0.006}$ | $\mathbf{0.0203 \pm 0.0011}$ | $\mathbf{0.0570 \pm 0.0006}$ | $\mathbf{5.177 \pm 0.036}$ |
| MAPE | ERM | $1.753 \pm 0.078\%$ | $13.615 \pm 0.165\%$ | $2.423 \pm 0.365\%$ | $13.861 \pm 0.152\%$ | $8.700 \pm 0.015\%$ |
| | mixup | $2.327 \pm 0.159\%$ | $13.534 \pm 0.125\%$ | $2.441 \pm 0.286\%$ | $14.306 \pm 0.048\%$ | $8.838 \pm 0.108\%$ |
| | Mani mixup | $1.842 \pm 0.114\%$ | $13.382 \pm 0.360\%$ | $2.475 \pm 0.346\%$ | $14.556 \pm 0.057\%$ | $8.955 \pm 0.082\%$ |
| | k-Mixup | $1.769 \pm 0.035\%$ | $13.173 \pm 0.139\%$ | $2.403 \pm 0.311\%$ | $14.134 \pm 0.134\%$ | $9.206 \pm 0.117\%$ |
| | Local Mixup | $2.290 \pm 0.101\%$ | $13.202 \pm 0.176\%$ | $2.341 \pm 0.229\%$ | $14.245 \pm 0.152\%$ | $9.313 \pm 0.115\%$ |
| | MixRL | $2.163 \pm 0.219\%$ | $13.298 \pm 0.182\%$ | $2.397 \pm 0.296\%$ | $14.417 \pm 0.203\%$ | $9.165 \pm 0.134\%$ |
| | **C-Mixup (Ours)** | $\mathbf{1.610 \pm 0.085\%}$ | $\mathbf{12.998 \pm 0.271\%}$ | $\mathbf{2.041 \pm 0.134\%}$ | $\mathbf{13.372 \pm 0.106\%}$ | $\mathbf{8.435 \pm 0.089\%}$ |

# D    Additional Experiments of Task Generalization

## D.1    Detailed Dataset Description

**ShapeNet1D.** We adopt the same preprocessing strategy to preprocess the ShapeNet1D dataset [5]. The ShapeNet1D dataset contains 27 categories with 60 objects per category. For each category, we randomly select 50 objects for meta-training and the rest ones are used for meta-testing. The model takes a $128 \times 128$ grey-scale image as the input, and the label is normalized to $[0, 10]$.

**PASCAL3D.** In PASCAL3D, we follow [18] to preprocess the dataset, where 50 and 15 categories are used for meta-training and meta-testing, respectively. The input image size and output label scale are same as ShapeNet1D.

## D.2    Hyperparameters.

We list the hyperparameters used in the experiments of ShapeNet1D and PASCAL3D in Table 5.

Table 5: Hyperparameters of task generalization experiments.

| Hyperparameters | ShapeNet1D | PASCAL3D |
|---|---|---|
| outer-loop learning rate | 0.0005 | 0.001 |
| inner-loop learning rate | 0.002 | 0.01 |
| # of inner-loop updates | 5 | 5 |
| $\alpha$ in Beta Dist. | 0.5 | 0.5 |
| batch size | 10 | 10 |
| support/query shot | 15 | 15 |
| max. iterations | 15,000 | 15,000 |

# E    Additional Experiments of Out-of-Distribution Robustness

## E.1    Detailed Dataset Description

We provide detailed descriptions for datasets that are used in the experiments of out-of-distribution robustness.

**RCF-MNIST.** The prefix "RCF" of RCF-MNIST means "Rotated-Colored-Fashion". To construct RCF-MNIST, assume the normalized RGB vector of red and blue is $[1, 0, 0]$ and $[0, 0, 1]$ and the normalized angle of rotation (i.e., label) for one image is $g \in [0, 1]$. In training set, we color 80% images with RGB value $[g, 0, 1 - g]$ and the rest images are colored with $[1 - g, 0, g]$. Hence, the color information is strongly spuriously correlated with the label in the training set.

Table 6: Spurious correlation analysis in RCF-MNIST. We list the test performance w/ and w/o distribution shifts.

| | w/o shift | w/ shift |
|---|---|---|
| RMSE $\downarrow$ | 0.111 | 0.162 |

In test set, we reverse spurious correlations to simulate distribution shift, where 80% and 20% images are colored with RGB values $[1 - g, 0, g]$ and $[g, 0, 1 - g]$, respectively. We further verify that the spurious correlation between color and label affects the performance. Here, we compare the performance of same test set with or without distribution shift. The results are reported in Table 6, where we observe that the subpopulation shift caused by spurious correlation does hurt the performance as expected.

**PovertyMap** is included in the WILDS benchmark [9], which contains satellite images from 23 African countries that can be used to predict the village-level real-valued asset wealth index. The input is a $224 \times 224$ multispectral LandSat satellite image with 8 channels, and the label is the real-valued asset wealth index. The domains of the images consist the country, urban and rural area information. This dataset includes 5 different cross validation folds, and all countries in these splits are disjoint to support the out-of-distribution setting. All experimental settings follow Koh et al. [9].

**Communities And Crimes (Crime)** is a tabular dataset combining socio-economic data from the 1990 US Census, law enforcement data from the 1990 US LEMAS survey, and crime data from the 1995 FBI UCR. The input features include 122 attributes that have some plausible connection to crime, such as the median family income and percent of officers assigned to drug units. The label attribute to be predicted is per capita violent crimes, which covers violent crimes including

murder, rape, robbery, assault and so on. All numeric features are normalized into the decimal range $0.00 \sim 1.00$ by equal-interval binning method, and the missing values are filled with the average values of the corresponding attributes. State identifications are used as the domain information, resulting to 46 domains in total. We split the dataset into training, validation and test sets with size 1,390, 231 and 373, while they contain 31, 6 and 9 disjoint domains, respectively.

**SkillCraft.** SkillCraft is a UCI tabular dataset [2] originated from a study that used video game telemetry data from real-time strategy (RTS) games to explore the development of expertise. Input x contains 17 player-related parameters in the game, such as the Cognition-Action-cycle variables and the Hotkey Usage variables. And the action latency in the game was considered as the label y. Missing data are filled by mean padding on each attribute. We use "League Index", which correspond to different levels of competitors, to be the identifier of domain. The dataset is split into training, validation and test sets with size 1878, 806, 711 and disjoint domain number 4, 1, 3, respectively.

**Drug-target Interactions (DTI).** Drug-target Interactions dataset [7] originated aims to predict the binding activity score between each small molecule and the corresponding target protein. The input features contain both drug and target protein information, which are represented by one-hot vectors. The output label is the binding activity score. The training and validation set are selected from 2013-2018, and test set is drawn from 2019-2020. We regard "Year" as the domain information.

### E.2 Hyperparameters

We list the hyperparameters for the experiments of out-of-distribution robustness in Table 7.

Table 7: Hyperparameter settings for the experiments of out-of-distribution robustness.

| Dataset | RCF-MNIST | PovertyMap | Crime | SkillCraft | DTI |
|---|---|---|---|---|---|
| Learning rate | 7e-5 | 1e-3 | 1e-3 | 1e-2 | 5e-5 |
| Weight decay | 0 | 0 | 0 | 0 | 0 |
| Scheduler | n/a | StepLR | n/a | n/a | n/a |
| Batch size | 64 | 64 | 16 | 32 | 64 |
| Type of mixup | ManiMix | mixup | ManiMix | mixup | ManiMix |
| Architecture | ResNet-18 | ResNet-50 | FCN3 | FCN3 | DeepDTA |
| Optimizer | Adam | Adam | Adam | Adam | Adam |
| Maximum Epoch | 30 | 50 | 200 | 100 | 20 |
| Bandwidth $\sigma$ | 0.2 | 0.5 | 1.0 | 5e-4 | 21.0 |
| $\alpha$ in Beta Dist. | 2.0 | 0.5 | 2.0 | 2.0 | 2.0 |

### E.3 Full Results

In Table 8, we report the full results of out-of-distribution robustness.

## F Additional Analysis of C-Mixup

### F.1 Additional Compatibility Analysis

In Table 9, we report the full results of compatibility analysis. Here, the performances on ERM, mixup, mixup+C-Mixup are also reported for comparison. In addition to the compatibility of C-Mixup, we also observe that some powerful inter-class mixup policies (e.g., PuzzleMix) improve the performance on part of regression tasks, e.g., RCF-MNIST. However, these approaches may also yield worse performances than ERM in other datasets, e.g., PovertyMap. Nevertheless, integrating C-Mixup on these mixup-based variants performs better than their vanilla versions, showing the compatibility and complementarity of C-Mixup to the existing mixup-based approaches in regression.

### F.2 Distance Metrics

In this section, we first discuss how to calculate the representation distance $d(h_i, h_j)$. Then, we provide complete analysis of distance metrics.

Table 8: Full results of out-of-distribution robustness. The standard deviations are calculated by 5-fold data split in PovertyMap [9], or over 3 seeds in other datasets.

| | RCF-MNIST (RMSE) | PovertyMap ($R$) | | Crime (RMSE) | |
| --- | --- | --- | --- | --- | --- |
| | Avg. ↓ | Avg. ↑ | Worst ↑ | Avg. ↓ | Worst ↓ |
| ERM | $0.162 \pm 0.003$ | $0.80 \pm 0.04$ | $0.50 \pm 0.07$ | $0.134 \pm 0.003$ | $0.173 \pm 0.009$ |
| IRM | $0.153 \pm 0.003$ | $0.77 \pm 0.05$ | $0.43 \pm 0.07$ | $0.127 \pm 0.001$ | $0.155 \pm 0.003$ |
| IB-IRM | $0.167 \pm 0.003$ | $0.78 \pm 0.05$ | $0.40 \pm 0.05$ | $0.127 \pm 0.002$ | $0.153 \pm 0.004$ |
| V-REx | $0.154 \pm 0.011$ | $\mathbf{0.83 \pm 0.02}$ | $0.48 \pm 0.03$ | $0.129 \pm 0.005$ | $0.157 \pm 0.007$ |
| CORAL | $0.163 \pm 0.016$ | $0.78 \pm 0.05$ | $0.44 \pm 0.06$ | $0.133 \pm 0.007$ | $0.166 \pm 0.015$ |
| GroupDRO | $0.232 \pm 0.016$ | $0.75 \pm 0.07$ | $0.39 \pm 0.06$ | $0.138 \pm 0.005$ | $0.168 \pm 0.009$ |
| Fish | $0.263 \pm 0.017$ | $0.80 \pm 0.02$ | $0.30 \pm 0.01$ | $0.128 \pm 0.000$ | $0.152 \pm 0.001$ |
| mixup | $0.176 \pm 0.003$ | $0.81 \pm 0.04$ | $0.46 \pm 0.03$ | $0.128 \pm 0.002$ | $0.154 \pm 0.001$ |
| **Ours** | $\mathbf{0.146 \pm 0.005}$ | $0.81 \pm 0.03$ | $\mathbf{0.53 \pm 0.07}$ | $\mathbf{0.123 \pm 0.000}$ | $\mathbf{0.146 \pm 0.002}$ |

| | n/a | SkillCraft (RMSE) | | DTI ($R$) | |
| --- | --- | --- | --- | --- | --- |
| | n/a | Avg. ↓ | Worst ↓ | Avg. ↑ | Worst ↑ |
| ERM | n/a | $5.887 \pm 0.362$ | $10.182 \pm 1.745$ | $0.464 \pm 0.014$ | $0.429 \pm 0.004$ |
| IRM | n/a | $5.937 \pm 0.254$ | $7.849 \pm 0.371$ | $0.478 \pm 0.007$ | $0.432 \pm 0.003$ |
| IB-IRM | n/a | $6.055 \pm 0.503$ | $7.650 \pm 0.653$ | $0.479 \pm 0.009$ | $0.435 \pm 0.007$ |
| V-REx | n/a | $6.059 \pm 0.429$ | $7.444 \pm 0.494$ | $0.485 \pm 0.009$ | $0.435 \pm 0.004$ |
| CORAL | n/a | $6.353 \pm 0.102$ | $8.272 \pm 0.436$ | $0.483 \pm 0.010$ | $0.432 \pm 0.005$ |
| GroupDRO | n/a | $6.155 \pm 0.537$ | $8.131 \pm 0.608$ | $0.442 \pm 0.043$ | $0.407 \pm 0.039$ |
| Fish | n/a | $6.356 \pm 0.201$ | $8.676 \pm 1.159$ | $0.470 \pm 0.022$ | $0.443 \pm 0.010$ |
| mixup | n/a | $5.764 \pm 0.618$ | $9.206 \pm 0.878$ | $0.465 \pm 0.004$ | $0.437 \pm 0.016$ |
| **Ours** | n/a | $\mathbf{5.201 \pm 0.059}$ | $\mathbf{7.362 \pm 0.244}$ | $\mathbf{0.498 \pm 0.008}$ | $\mathbf{0.458 \pm 0.004}$ |

Table 9: Full results (performance with standard deviation) of compatibility analysis.

| Model | | RCF-MNIST | PovertyMap |
| --- | --- | --- | --- |
| | | RMSE ↓ | Worst $R$ ↑ |
| ERM | | $0.162 \pm 0.003$ | $0.50 \pm 0.07$ |
| mixup | | $0.176 \pm 0.003$ | $0.46 \pm 0.03$ |
| | +C-Mixup | $\mathbf{0.146 \pm 0.005}$ | $\mathbf{0.53 \pm 0.07}$ |
| CutMix | | $0.194 \pm 0.010$ | $0.49 \pm 0.05$ |
| | +C-Mixup | $\mathbf{0.186 \pm 0.013}$ | $\mathbf{0.52 \pm 0.06}$ |
| PuzzleMix | | $0.159 \pm 0.004$ | $0.47 \pm 0.03$ |
| | +C-Mixup | $\mathbf{0.150 \pm 0.012}$ | $\mathbf{0.50 \pm 0.04}$ |
| AutoMix | | $0.152 \pm 0.021$ | $0.49 \pm 0.07$ |
| | +C-Mixup | $\mathbf{0.146 \pm 0.009}$ | $\mathbf{0.53 \pm 0.07}$ |

**Measuring Representation Distance.** In this paper, we adopt a two-stage training process for each iteration. In the first stage, we feed the data into the current backbone and get hidden representations $h$, which is used to calculate the example distance, i.e., $d(h_i, h_j)$. In the second stage, we apply C-Mixup with representation distance.

**Complete Analysis of Distance Metrics** We report full results of the analysis of distance metrics in Table 10. In addition to the existing analysis, we conduct one analysis by changing how to calculate the distance between low-dimensional hidden representations, where we compare the Euclidean distance and the cosine distance. We observe that C-Mixup performs better than using both Euclidean and cosine distances to measure the similarity between low-dimensional representations, corroborating the effectiveness of C-Mixup.

## F.3   Additional Hyperparameter Sensitivity

In this section, we first provide more experiments for bandwidth analysis. Then, we conduct experiments to show the effect of hyperparameter $\alpha$ in Beta distribution, i.e., $\text{Beta}(\alpha, \alpha)$.

Table 10: Full results (performance with standard deviation) of different distance metrics.

| Model | Exchange-Rate | ShapeNet1D | DTI |
|---|---|---|---|
| | RMSE ↓ | MSE ↓ | Avg. $R$ ↑ |
| ERM/MAML | $0.0236 \pm 0.0031$ | $4.698 \pm 0.079$ | $0.464 \pm 0.014$ |
| mixup/MetaMix | $0.0239 \pm 0.0027$ | $4.275 \pm 0.082$ | $0.465 \pm 0.004$ |
| $d(x_i, x_j)$ | $0.0212 \pm 0.0014$ | $4.539 \pm 0.082$ | $0.478 \pm 0.003$ |
| $d(x_i \oplus y_i, x_j \oplus y_j)$ | $0.0212 \pm 0.0009$ | $4.395 \pm 0.085$ | $0.484 \pm 0.002$ |
| $d(h_i, h_j)$ (Euclidean distance) | $0.0213 \pm 0.0006$ | $4.202 \pm 0.078$ | $0.483 \pm 0.001$ |
| $d(h_i, h_j)$ (Cosine distance) | $0.0209 \pm 0.0012$ | $4.411 \pm 0.081$ | $0.477 \pm 0.004$ |
| $d(h_i \oplus y_i, h_j \oplus y_j)$ | $0.0208 \pm 0.0016$ | $4.176 \pm 0.077$ | $0.487 \pm 0.001$ |
| $d(y_i, y_j)$ **(C-Mixup)** | $\mathbf{0.0203 \pm 0.0011}$ | $\mathbf{4.024 \pm 0.081}$ | $\mathbf{0.498 \pm 0.008}$ |

### F.3.1 Additional Bandwidth Analysis

We illustrate the bandwidth analysis for additional four datasets in Figure 3, including Airfoil, NO2, PovertyMap, SkillCraft. The results corroborate our finding in the main paper that C-Mixup yields a good model in a relative wide range of bandwidth, reducing the efforts to tune the bandwidth $\sigma$ for every specific dataset.

According to our empirical results, we conclude that roughly tuning the bandwidth in the range [0.01, 0.1, 1, 10, 100] is sufficient to get a relatively satisfying performance. To get the optimal bandwidth, we suggest to perform grid search.

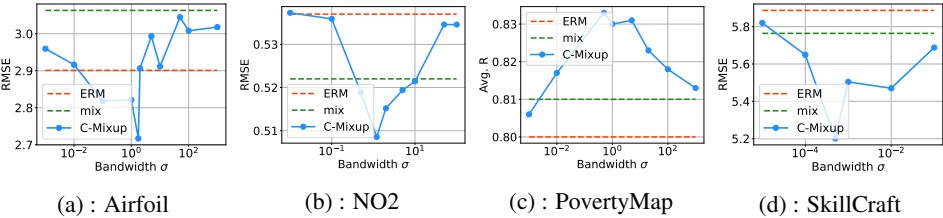

(a) : Airfoil      (b) : NO2      (c) : PovertyMap      (d) : SkillCraft

Figure 3: Additional robustness analysis of bandwidth

### F.3.2 Effect of Shape Parameter $\alpha$ in Beta Distribution

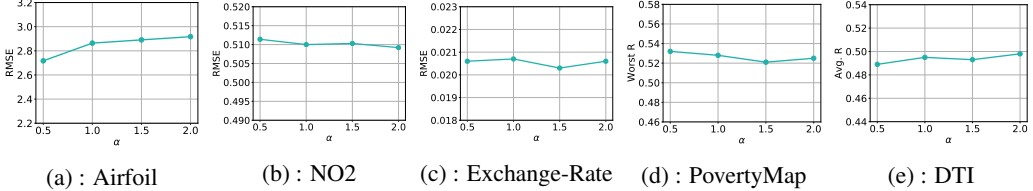

(a) : Airfoil      (b) : NO2      (c) : Exchange-Rate      (d) : PovertyMap      (e) : DTI

Figure 4: Robustness analysis of $\alpha$ in Beta distribution

Finally, we analyze the effect of shape parameter $\alpha$ in the Beta distribution. The results of five datasets are illustrated in Figure 4, including Airfoil, NO2, Exchange-Rate, PovertyMap, and DTI. We observe that the performance is relatively stable with the change of $\alpha$, indicating the robustness of C-Mixup to the shape of Beta distribution.

### F.4 Robustness Analysis to Label Noise

We conduct experiments to investigate the robustness of C-Mixup to label noise. Specifically, we inject Gaussian noises into labels for all training examples. For each dataset, the noise is set as 30% of the standard deviation of the corresponding original labels, where adding noise significantly

degrades the performance compared to that with clean data. The results and the corresponding noise distributions on three datasets – Exchange-Rate, ShapeNet1D, DTI are reported in Table 11. According to Table 11, C-Mixup still improves the performance over ERM and vanilla mixup, showing its robustness to label noise.

Table 11: Robustness analysis to label noise.

| Model | Exchange-Rate | ShapeNet1D | DTI |
|---|---|---|---|
| | RMSE $\downarrow$ | MSE $\downarrow$ | Avg. $R \uparrow$ |
| Noise Type | $\mathcal{N}(0, 1.18 \times 10^{-3})$ | $\mathcal{N}(0, 0.874)$ | $\mathcal{N}(0, 7.59 \times 10^{-3})$ |
| ERM/MAML | $0.0381 \pm 0.0014$ | $5.553 \pm 0.098$ | $0.334 \pm 0.018$ |
| mixup/MetaMix | $0.0375 \pm 0.0017$ | $5.329 \pm 0.101$ | $0.307 \pm 0.021$ |
| **C-Mixup** | $\mathbf{0.0360 \pm 0.0013}$ | $\mathbf{5.185 \pm 0.096}$ | $\mathbf{0.356 \pm 0.013}$ |