# OpenReview forum: "C-Mixup: Improving Generalization in Regression"
_NeurIPS.cc/2022/Conference — NeurIPS 2022 Accept_

### Official Review · Reviewer_VT7i · 2022-07-09

**Rating:** 6
**Confidence:** 4
**Soundness:** 3 good
**Presentation:** 3 good
**Contribution:** 2 fair

**Summary:**

In this work, the authors propose to use mixup to improve generalization in regression tasks. They argue that when mixup is directly using in regression, it may generate arbitrary labels which are incorrect as the linear assumption may not hold. The authors aim to overcome this drawback by adjusting the sampling probability based on the similarity of the labels. They further show that this label similarity obtains smaller mean square error and also improves out-of-distribution robustness. Interpolating examples with similar labels mitigates the effect of domain-specific information and pushes invariant representations. The authors evaluate their work on several benchmarks and show its effectiveness on in-distribution generalization, task generalization and out-of-distribution robustness.

**Questions:**

#### **Weakness**/ **Questions**

I appreciate the authors efforts in developing an interesting idea. However, I have the following concerns/ queries which I would like the authors to answer:
- I quite did not understand the difference between sampling probability and the interpolation factor $\lambda$ typically used in mixup papers.
- In L133, the authors say that its meaningful to use $d(i,j) = ||y_i - y_j||^2$. However, what is the dimension of $y$? is it a one-hot vector. This is quite unclear again. Also instead of using the $L_2$ distance, how about using cosine similarity in lower-dimensional embeddings instead of $L_2$ distance in the label space?
- In the original mixup paper [62], the authors show that interpolating examples from semantically similar manifolds leads to sub-optimal performance and the best performance is achieved by randomly mixing samples from the data manifold. However, in MixReg, the authors claim that mixup samples from semantically similar data manifold results in the best performance. Could the authors please explain its significance?
- Continuing from the above point, interpolating between semantically similar points in the data manifold could lead to a problem of manifold intrusion [1] - Given two examples with different class labels, the interpolated example may actually lie in a region associated with a third class in the feature space. Im curious to know if the authors face this problem in MixReg, if not, what is being done to avoid manifold intrusion.
[1] Guo et al., Mixup as locally linear out-of-manifold regularization. In AAAI, 2019.

- Could the authors evaluate their work on large scale tasks such as semantic segmentation which is also a pixel-wise regression task on Pascal-VOC or MS-COCO? How does one apply mixup to such kind of tasks in general?
- The authors have missed citing several state-of-the-art mixup works such as PuzzleMix [2], Co-Mixup [3], SaliencyMix [4], AlignMixup [5], StyleMix [6], StyleCutMix [6],  AutoMix [7] etc. It would be nice to discuss these papers in the related work section.

[2] Kim et al., Puzzle mix: Exploiting saliency and local statistics for optimal mixup. ICML 2020.
[3] Kim et al., Co-mixup: Saliency guided joint mixup with supermodular diversity. ICLR 2021.
[4] Uddin et al. Saliencymix: A saliency guided data augmentation strategy for better regularization. ICML 2021.
[5] Venkataramanan et al., AlignMixup: Improving Representations By Interpolating Aligned Features. CVPR 2022.
[6] Hong et al., Stylemix: Separating content and style for enhanced data augmentation. CVPR, 2021.
[7]  Zhu et al., Automix: Mixup networks for sample interpolation via cooperative barycenter learning. ECCV, 2020


**Limitations:**

Yes

**Strengths And Weaknesses:**

#### **Strengths**
The following are some strengths of the work
- The authors argument that when mixup is directly applied to regression task generates incorrect labels due to the failure of the linear assumption rule is quite interesting. Its also quite interesting that mixup in regression tasks has been understudied and the authors aim to bridge this gap.
- The experimental analysis is quite exhaustive and the authors show the performance of MixReg on many different benchmarks and in-distribution generalization, task generalization and out-of-distribution robustness.
- The paper is also quite clear to read and is well presented.

---

> ### Author Response · Authors · 2022-08-02
> **Response to Reviewer VT7i (1/3)**
>
> Thank you for your constructive comments and suggestions. We have revised our paper according to your comments. We respond to your questions below and would appreciate it if you could let us know if our response addresses your concerns.
>
> > **Q1**: I quite did not understand the difference between sampling probability and the interpolation factor λ typically used in mixup papers.
>
> **A1**: The entire mixup process includes three stages:
> - Stage I: Sample two instances ($x_i$, $y_i$), ($x_j$, $y_j$) from the training set
> - Stage II: Sample the interpolation factor \lambda from the Beta distribution Beta(\alpha, \alpha)
> - Stage III: Mixing the sampled instances with interpolation factor \lambda according to the following mixing formulation:
>
> $x_{mix}=\lambda x_i + (1-\lambda) x_j, y_{mix}=\lambda y_i + (1-\lambda) y_j, \lambda \sim \mathrm{Beta}(\alpha, \alpha)$.
>
> In the original mixup, the interpolation factor $\lambda$ sampled in stage II controls how to mix these two instances. mixReg instead manipulates stage I and pairs with closer labels are more likely to be sampled. In the revised version, we have added the above discussion in Appendix A.4 and mentioned it in Section 3.1 of the main paper.
>
> ---
>
>
> > **Q2**: In L133, the authors say that its meaningful to use d(i,j)=||yi−yj||2. However, what is the dimension of y? is it a one-hot vector. This is quite unclear again.
>
> **A2**: y is a vector with continuous values. We’ve clarified it in Line 129 of Section 3.1 in the revised paper. As mentioned in Line 130-132, the dimension of y is typically smaller compared to the input feature dimension, and sometimes equal to 1 (e.g., PovertyMap, DTI).
>
> ---
>
> >  **Q3**: Instead of using the L2 distance, how about using cosine similarity in lower-dimensional embeddings instead of L2 distance in the label space?
>
> **A3**: We conduct new experiments using cosine similarity in low-dimensional embeddings (i.e., hidden representations) and report the results below in Table R1. We also report the performance of mixReg and using L2 distance in lower-dimensional embeddings (copied from our analysis II in Section 4.4 of the revised paper) for comparison.
>
> **Table R1**: Comparison between label similarity and cosine representation similarity. $\downarrow$ means the smaller the better and $\uparrow$ means the larger the better.
>
> | Model                       | Exchange-Rate | ShapeNet1D  | DTI  |
> |-----------------------------|---------------|------------|-------|
> |     |  RMSE $\downarrow$ |  MSE $\downarrow$  |  Avg. R $\uparrow$ |
> | Euclidean distance on low-dim representation | 0.0213 $\pm$ 0.0006        | 4.202 $\pm$ 0.078      | 0.483 $\pm$ 0.001 |
> | Cosine distance on low-dim representation    | 0.0209 $\pm$ 0.0012        | 4.411 $\pm$ 0.081      | 0.477 $\pm$ 0.004 |
> | **mixReg (ours)**                      | **0.0203 $\pm$ 0.0011**        | **4.024 $\pm$ 0.081**      | **0.498 $\pm$ 0.008** |
>
> According to the results, using label similarity (i.e., mixReg) still performs better than using both L2 distance and cosine similarity in lower-dimensional hidden representations, corroborating the efficacy of mixReg. We have added the comparison between mixReg and using cosine similarity on hidden representations in Appendix F.2.
>
> ---
>
> > **Q4**: In the original mixup paper [62], the authors show that interpolating examples from semantically similar manifolds leads to sub-optimal performance and the best performance is achieved by randomly mixing samples from the data manifold. However, in MixReg, the authors claim that mixup samples from semantically similar data manifold results in the best performance. Could the authors please explain its significance?
>
> **A4**: The original mixup paper focuses on classification datasets. As discussed in Line 31-35 and Figure 1, in regression, randomly mixing examples may be easier to generate semantically wrong labels compared to classification. Intuitively, linearly mixing one-hot labels in classification is easy to generate semantically meaningful artificial labels, where the mixed label represents the probabilities of mixed examples to some extent. While in regression, the mixed labels may be semantically meaningless (e.g., pairs 2 and 3 in Figure 1) and more significantly affect the performance. By mixing examples with closer labels, mixReg mitigates the influence of semantically wrong labels and improves the in-distribution and task generalization in regression. Additionally, mixReg further shows its superiority in improving out-of-distribution robustness in regression, which is not discussed in the original mixup paper. We’ve added the above discussion in Appendix A.4 of the revised paper. In our submission, we justified these advantages of mixReg over vanilla mixup from both theoretical and empirical perspectives.

---

> > ### Author Response · Authors · 2022-08-02
> > **Response to Reviewer VT7i (2/3)**
> >
> > > **Q5**: Continuing from the above point, interpolating between semantically similar points in the data manifold could lead to a problem of manifold intrusion - Given two examples with different class labels, the interpolated example may actually lie in a region associated with a third class in the feature space. Im curious to know if the authors face this problem in MixReg, if not, what is being done to avoid manifold intrusion.
> >
> >
> > **A5**: MixReg specifically aims to interpolate examples with closer labels, which mitigates the effects of manifold intrusion to some extent. Here, we use one example to illustrate it:
> >
> > In this example, we aim to predict the angle of rotation of an object. Given different colors of background that take a large portion of pixels in an image, we consider two pairs of instances: pair 1 [(angle=50, color=yellow), (angle=50.5, color=blue)],  pair 2 [(angle=50, color=yellow), (angle=100, color=yellow)]. If we consider the input feature similarity (similarity in x), examples in pair 2 will have a much smaller distance, and therefore will be mixed with a higher probability than pair 1. In this case, the manifold intrusion happens: we obtain a mixed image with a position that has no semantic meaning. In contrast, if we consider the label similarity (similarity in y), then examples in pair 1 will have a smaller distance and be mixed with a higher probability, leading to a new image with an angle between 50 and 50.5 with a different color. In this example, mixReg alleviates the manifold intrusion problem.
> >
> > Despite mixReg's ability to mitigate manifold intrusion, we believe there is still a lot of room for further investigation of manifold intrusion in regression and plan to leave it for future work, which is added in our discussion of limitations, i.e., Appendix G of the revised paper.
> >
> > ---
> >
> > > **Q6**: Could the authors evaluate their work on large scale tasks such as semantic segmentation which is also a pixel-wise regression task on Pascal-VOC or MS-COCO? How does one apply mixup to such kind of tasks in general?
> >
> > **A6**: The paper already evaluated mixReg on large-scale datasets, such as Echocardiogram Video [Ouyang et al., Nature 2020], PovertyMap [Yeh et al., Nature Communication 2020], and DTI [Huang et al., NeurIPS 2021]. Specifically, Echocardiogram Video is one of the largest public labeled medical video datasets for cardiac function assessments, including 10,030 apical-4-chamber echocardiography videos. PovertyMap is a regression dataset in the Wilds benchmark [Koh et al. ICML 2021], aiming to estimate global-scale poverty with satellite images. Drug-target Interaction is the only domain shift task in Therapeutics Data Commons — one of the largest public drug discovery benchmarks. We’ve revised the last paragraph of the introduction to highlight these datasets.
> >
> > In terms of semantic segmentation, mixup has not been widely used in semantic segmentation to our knowledge. We leave this task for future work (Appendix G of the revised paper).

---

> > > ### Author Response · Authors · 2022-08-02
> > > **Response to Reviewer VT7i (3/3)**
> > >
> > > > **Q7**: The authors have missed citing several state-of-the-art mixup works such as PuzzleMix [2], Co-Mixup [3], SaliencyMix [4], AlignMixup [5], StyleMix [6], StyleCutMix [6], AutoMix [7] etc. It would be nice to discuss these papers in the related work section.
> > >
> > > **A7**: Thank you for pointing out these relevant references. We have added and discussed these papers in the related work section. We would also like to point out that mixReg is a complementary approach to these mixup variants, where mixReg changes the probabilities of sampling mixing pairs instead of changing the way to mixing. We further conduct compatibility analysis of mixReg on three representative large-scale regression datasets: Exchange-Rate (time-series prediction), PovertyMap (image regression), and Echo (video regression). We report the results in Table R2, where we basically inject mixReg to three representative mixup variants (PuzzleMix, CutMix, AutoMix). The results validate the compatibility of mixReg with these prior methods. We’ve added these new experimental results and discussion of compatibility in Section 4.4 (analysis I) and Appendix F.1 of the revised paper.
> > >
> > > **Table R2**: Compatibility analysis of mixReg. $\downarrow$: the smaller the better; $\uparrow$: the larger the better.
> > >
> > > | Model     |         | Exchange-Rate | Echo   | PovertyMap |
> > > |-----------|---------|---------------|-------|------------|
> > > |      |         |  RMSE $\downarrow$ |  RMSE $\downarrow$  |  Worst R $\uparrow$ |
> > > | CutMix    |         | 0.0264   $\pm$  0.0049   | 5.405 $\pm$ 0.069 | 0.49 $\pm$ 0.05      |
> > > |           | **+mixReg** | **0.0240 $\pm$ 0.0021**        | **5.161 $\pm$ 0.062** | **0.52 $\pm$ 0.06**       |
> > > | PuzzleMix |         | 0.0254 $\pm$ 0.0027       | 5.368 $\pm$ 0.095 | 0.47 $\pm$ 0.03      |
> > > |           |  **+mixReg** | **0.0233 $\pm$ 0.0012**        | **5.206 $\pm$ 0.063** | **0.50 $\pm$ 0.04**       |
> > > | AutoMix   |         | 0.0242 $\pm$ 0.0033       | 5.525 $\pm$ 0.055 | 0.50 $\pm$ 0.06       |
> > > |           |  **+mixReg** | **0.0228 $\pm$ 0.0014**       | **5.239 $\pm$ 0.037** | **0.53 $\pm$ 0.06**       |
> > >
> > > ---
> > >
> > >
> > >
> > > **Reference**
> > >
> > > [Ouyang et al., Nature 2020] Ouyang, David, Bryan He, Amirata Ghorbani, Neal Yuan, Joseph Ebinger, Curtis P. Langlotz, Paul A. Heidenreich et al. "Video-based AI for beat-to-beat assessment of cardiac function." Nature 580, no. 7802 (2020): 252-256.
> > >
> > > [Yeh et al. Nature Communication 2020] Yeh, Christopher, Anthony Perez, Anne Driscoll, George Azzari, Zhongyi Tang, David Lobell, Stefano Ermon, and Marshall Burke. "Using publicly available satellite imagery and deep learning to understand economic well-being in Africa." Nature communications 11, no. 1 (2020): 1-11.
> > >
> > > [Koh et al. ICML 2021] Koh, Pang Wei, Shiori Sagawa, Henrik Marklund, Sang Michael Xie, Marvin Zhang, Akshay Balsubramani, Weihua Hu et al. "Wilds: A benchmark of in-the-wild distribution shifts." In International Conference on Machine Learning, pp. 5637-5664. PMLR, 2021.
> > >
> > > [Huang et al. NeurIPS 2021] Huang, Kexin, Tianfan Fu, Wenhao Gao, Yue Zhao, Yusuf Roohani, Jure Leskovec, Connor W. Coley, Cao Xiao, Jimeng Sun, and Marinka Zitnik. "Therapeutics Data Commons: Machine Learning Datasets and Tasks for Drug Discovery and Development." In NeurIPS Datasets and Benchmarks. 2021.

---

> > > > ### Author Response · Authors · 2022-08-05
> > > > **We would like to hear back from Reviewer VT7i**
> > > >
> > > > Hi reviewer VT7i,
> > > >
> > > > We would like to follow up to see if the response addresses your concerns or if you have any further questions. We would really appreciate the opportunity to discuss this further if our response has not already addressed your concerns. Thank you again!

---

> > > > ### Comment · Reviewer_VT7i · 2022-08-05
> > > > **Response to the authors**
> > > >
> > > > I really appreciate the very detailed explanation for each of my queries given by the authors. The authors have answered most of my questions and their revision to the paper makes it much better now. Im willing to *raise my rating to 6*.
> > > >
> > > > I still have one follow-up question and  would appreciate if the authors could share their thoughts on them
> > > > - for Q1, its clear to me now the difference between sampling probability and $\lambda$. But from L114 of the paper, the authors say that they "introduces a symmetric Gaussian kernel in order to calculate the sampling probability" to form semantically closer pairs. I might have misunderstood this, but do the authors encounter the problem of having noisy pairs?

---

> > > > > ### Author Response · Authors · 2022-08-05
> > > > > **Response to Reviewer VT7i -- Follow-up Question**
> > > > >
> > > > > Dear Reviewer VT7i,
> > > > >
> > > > > We are happy to see our response addresses your questions. Thank you for raising your rating.
> > > > >
> > > > > In terms of your follow-up question, our understanding is that the noisy pairs are caused by noisy labels in the training set. Please kindly correct us if our understanding is wrong or if you have any follow-up questions.
> > > > >
> > > > > If it is correct, since Reviewer t9o8 also asked it in the initial review, we did conduct experiments to investigate the robustness of mixReg to label noise during the rebuttal period, which was also added in Appendix F.4 of the revised paper. Concretely, we injected Gaussian noises into the labels. The noise is set as 30% of the standard deviation among the corresponding original labels, where adding noise significantly degrades the performance compared to clean data. In Table R2, we report the results and the corresponding noise distributions on Exchange-Rate, ShapeNet1D, and DTI, respectively.
> > > > >
> > > > > **Table R2**: Robustness analysis to label noise. ($\downarrow$ denotes the smaller the better; $\uparrow$ denotes the larger the better)
> > > > >
> > > > > | Model           | Exchange-Rate | ShapeNet1D   | DTI |
> > > > > |---------------|---------------|-------|------------|
> > > > > |              |  RMSE $\downarrow$ |  MSE $\downarrow$  |  Avg. R $\uparrow$ |
> > > > > | Noise Type | $\mathcal{N}(0, 1.18\times10^{-3})$ | $\mathcal{N}(0, 0.874)$ | $\mathcal{N}(0, 7.59\times10^{-3})$ |
> > > > > | ERM/MAML | 0.0381 $\pm$ 0.0014        | 5.553 $\pm$ 0.098      | 0.334 $\pm$ 0.018 |
> > > > > |  mixup/MetaMix   | 0.0375 $\pm$ 0.0017        | 5.329 $\pm$ 0.101      | 0.307 $\pm$ 0.021 |
> > > > > | **mixReg (ours)**                      | **0.0360 $\pm$ 0.0013**        | **5.185 $\pm$ 0.096**      | **0.356 $\pm$ 0.013** |
> > > > >
> > > > > According to Table R2, we observe that mixReg still improves the performance over ERM and vanilla mixup even with the addition of label noise, showing its effectiveness and robustness to label noise.

---

### Official Review · Reviewer_13Sq · 2022-07-11

**Rating:** 6
**Confidence:** 4
**Soundness:** 3 good
**Presentation:** 3 good
**Contribution:** 3 good

**Summary:**

This paper proposed a mixup algorithm (mixReg) to improve generalization on regression tasks. Different from mixup methods for classification tasks, the proposed method adjusts the sampling probability based on the similarity of labels. Theoretical analysis shows that the proposed mixReg can achieve a lower mean square error and improve in-distribution generalization, task generalization, and out-of-distribution robustness. Experiments on eleven datasets prove the improvements of the proposed mixReg.

**Questions:**

Why mixReg work? The proposed mixReg only takes the intra-cluster (semantically similar) samples to perform vanilla mixup. Although the theoretical analysis proves the superior of mixReg upon vanilla mixup (and its variants) and mixup with input-similarity, the compared methods usually yield worse performances than ERM. The cause of bad performances of vanilla mixup might be the unreliable inter-cluster samples, and mixReg improves the mixup performance by removing these samples (similar to AdaMixup [2]). However, mixReg might degenerate to ERM with the large sample size N and the small bandwidth. What if we have a better inter-class mixup policy like PuzzleMix [3] and AutoMix [4]? We might transform regression tasks into classification tasks by splitting the regression targets into C bins and training the classifier to generate more reliable inter-class mixup samples [3, 4].

[1] CutMix: Regularization Strategy to Train Strong Classifiers with Localizable Features. In ICCV, 2019.

[2] MixUp as Locally Linear Out-Of-Manifold Regularization. In AAAI, 2019.

[3] Puzzle Mix: Exploiting Saliency and Local Statistics for Optimal Mixup. In ICML, 2020.

[4] AutoMix: Unveiling the Power of Mixup for Stronger Classifiers. In ECCV, 2022.


================ Post-rebuttal ================

Since the author has addressed my concerns and updated the revision according to my comments during the rebuttal period, I change my score from 5 to 6.

**Limitations:**

The author discusses the limitations of this paper in the appendix, which might be solved by further experiments. I do not find any negative social impact of this paper.

**Strengths And Weaknesses:**

### Strengths

(1) This paper studies an interesting problem that applies mixup augmentations to general regression tasks. Since data augmentations in regression tasks are rarely studied, the proposed mixReg is the first work that introduces the general regression augmentation strategy.

(2) The proposed method in three sceneries is well supported by theoretical analysis and proofs.

(3) Extensive experiments verify the effectiveness of the proposed mixReg from three aspects.

### Weakness

(1) Details of selecting mixup samples according to Eq. (6) are vague. There might be some thresholds to determine the intra-cluster samples based on Eq. (6). Meanwhile, the way to determine the bandwidth hyper-parameter is not clear. It is better to provide an empirical range of bandwidths for easy implementation in various scenarios.

(2) Although the author performs extensive experiments on many datasets, more mixup methods with various mixing policies should be compared. For example, CutMix [1] can be performed on 2D images and 1D time series (replacing the segment). See questions for more details.

---

> ### Author Response · Authors · 2022-08-02
> **Response to Reviewer 13Sq**
>
> Thank you for reviewing our paper and for your valuable feedback. Below, we address your concerns point by point and we’ve revised our paper according to your suggestions. We would appreciate it if you could let us know whether your concerns are addressed by our response.
>
> > **Q1**: Details of selecting mixup samples according to Eq. (6) are vague. There might be some thresholds to determine the intra-cluster samples based on Eq. (6).
>
> **A1**: Given an example ($x_i$, $y_i$), we did not set a threshold to select another example ($x_j$, $y_j$) in addition to Eq. (6). As shown in Line 6 of Algorithm 1, all examples conceptually have probabilities to be sampled as ($x_j$, $y_j$), but label closer examples have higher probabilities. The sampling distribution is controlled by the bandwidth $\sigma$. We clarified it in Line 136-138 of the revised paper.
>
> ---
>
> > **Q2**: the way to determine the bandwidth hyper-parameter is not clear.
>
> **A2**: As mentioned in the first paragraph of Section 4 of the revised paper, we determine the bandwidth $\sigma$ by performing grid search and applying cross-validation. As shown in Section 4.4 (analysis III) and Appendix F.3.1 of the revised paper, mixReg yields a good model for a wide range of bandwidths. Here, we recommend practitioners to try [0.01, 0.1, 1, 10, 100] if the computational resources are limited, which typically brings relatively satisfactory performance. We have revised Appendix C.2 and Appendix F.3.1 to highlight this guidance.
>
> ---
>
> > **Q3**: Although the author performs extensive experiments on many datasets, more mixup methods with various mixing policies should be compared.
>
> **A3**: Our proposed approach is a complementary method to the original mixup and its variants (e.g., CutMix, AutoMix, PuzzleMix). In mixReg, we basically change the sampling probability of mixing pairs, where examples with closer labels are more likely to be mixed. Other mixup variants (e.g., CutMix, AutoMix, PuzzleMix) instead focus on how to interpolate two examples. To further evaluate the compatibility of mixReg, we run new experiments on Exchange-Rate (time-series prediction), PovertyMap (image regression), and Echo (video regression) by incorporating mixReg with three representative mixup variants. We report the results in Table R1.
>
> **Table R1**: Compatibility analysis of mixReg. $\downarrow$: the smaller the better; $\uparrow$: the larger the better. The performances on ERM, mixup, mixup+mixReg are also reported for comparison.
>
> | Model     |         | Exchange-Rate | Echo   | PovertyMap |
> |-----------|---------|---------------|-------|------------|
> |      |         |  RMSE $\downarrow$ |  RMSE $\downarrow$  |  Worst R $\uparrow$ |
> | ERM    |         | 0.0236 $\pm$ 0.0031   | 5.402 $\pm$ 0.024 | 0.50 $\pm$ 0.07      |
> | mixup    |         | 0.0242 $\pm$ 0.0043  | 5.393 $\pm$ 0.040 | 0.46 $\pm$ 0.03     |
> |    |  **+mixReg**        | **0.0203 $\pm$  0.0011**   | **5.177 $\pm$ 0.036** | **0.51 $\pm$ 0.07**      |
> | CutMix    |         | 0.0264   $\pm$  0.0049   | 5.405 $\pm$ 0.069 | 0.49 $\pm$ 0.05      |
> |           | **+mixReg** | **0.0240 $\pm$ 0.0021**        | **5.161 $\pm$ 0.062** | **0.52 $\pm$ 0.06**       |
> | PuzzleMix |         | 0.0254 $\pm$ 0.0027       | 5.368 $\pm$ 0.095 | 0.47 $\pm$ 0.03      |
> |           |  **+mixReg** | **0.0233 $\pm$ 0.0012**        | **5.206 $\pm$ 0.063** | **0.50 $\pm$ 0.04**       |
> | AutoMix   |         | 0.0242 $\pm$ 0.0033       | 5.525 $\pm$ 0.055 | 0.50 $\pm$ 0.06       |
> |           |  **+mixReg** | **0.0228 $\pm$ 0.0014**       | **5.239 $\pm$ 0.037** | **0.53 $\pm$ 0.06**       |
>
>
> The results indicate that (1) compared to mixup, some powerful inter-class mixup policies (e.g., PuzzleMix) improve the performance on part of regression tasks, e.g., Echo. These approaches may also yield worse performances than ERM in other datasets, e.g., Exchange-Rate; (2) integrating mixReg on these mixup-based variants performs better than their vanilla versions, showing the compatibility and complementarity of mixReg to the existing mixup-based approaches in regression. We have revised our paper to include the compatibility analysis of mixReg (see analysis I in Section 4.4 and Appendix F.1).

---

> > ### Author Response · Authors · 2022-08-05
> > **We would like to hear back from Reviewer 13Sq**
> >
> > Hi Reviewer 13Sq,
> >
> > We would like to follow up to see if our response addresses your concerns or if you have any further questions. We would really appreciate the opportunity to discuss this further if our response has not already addressed your concerns. Thank you again!

---

> > ### Comment · Reviewer_13Sq · 2022-08-05
> > **Response to Authors**
> >
> > Thanks for the detailed rebuttal comments, which have conducted comparison experiments based on more mixup variants and addressed most of my concerns. The revision of the paper has provided more experiments and additional details on hyper-parameter tuning. I'm willing to raise my rating to 6.

---

> > > ### Author Response · Authors · 2022-08-05
> > > **Response to Reviewer t9o8**
> > >
> > > Dear Reviewer 13Sq,
> > >
> > > Thank you for your response, we are happy to see that our response address most of your concerns. Thanks again for your valuable comments to help us improve our paper and for raising your rating.

---

> > > > ### Comment · Reviewer_13Sq · 2022-08-07
> > > > **Update Response to Authors**
> > > >
> > > > Thanks for your respones and I'm glad to see the manuscript being improved in the revision. It will be better if the authors could cite several recently published state-of-the-art mixup works, e.g., Saliency Grafting [1], TransMix [2], AutoMix [3], TokenMix [4], etc. Since the proposed MixReg focuses on selecting proper mixing samples for regression tasks, it would be better to discuss the relationship with the mainstream mixup methods in classification tasks. Currently proposed methods improve the sample mixing policy (e.g., PuzzleMix, Co-Mixup, AutoMix [3]) or the label mixing policy (e.g., Saliency Grafting [1], TransMix [2], TokenMix [4]) with saliency or attention information.
> > > >
> > > > [1] Park, et al. Saliency Grafting: Innocuous Attribution-Guided Mixup with Calibrated Label Mixing. AAAI, 2022.
> > > >
> > > > [2] Chen, et al. TransMix: Attend to Mix for Vision Transformers. CVPR, 2022.
> > > >
> > > > [3] Liu, et al. AutoMix: Unveiling the Power of Mixup for Stronger Classifiers. ECCV, 2022.
> > > >
> > > > [4] Liu, et al. TokenMix: Rethinking Image Mixing for Data Augmentation in Vision Transformers. ECCV, 2022.

---

> > > > > ### Author Response · Authors · 2022-08-08
> > > > > **Additional Response to Reviewer 13Sq**
> > > > >
> > > > > Hi Reviewer 13Sq,
> > > > >
> > > > > Thank you for pointing out these recent works. We have cited Saliency Grafting [1], TransMix [2], AutoMix [3], TokenMix [4] in the updated version. Additionally, we added some discussion between mixReg and other classification-based mixup variants in both Line 382-385 (related work) and Appendix 4.1. As you mentioned, mixReg focuses on how to select mixing pairs in regression, while other approaches change the policy to mixing.

---

> > > > > > ### Comment · Reviewer_13Sq · 2022-08-08
> > > > > > **Advice on Citation**
> > > > > >
> > > > > > Thanks for your quick response. You may have confused two Automix articles. Automix [1] in ECCV'2020 generates mixup samples by generative methods, while Automix [2] in ECCV'2022 optimizes the mixup generation and mixup classification together in a closed loop to further improve PuzzleMix. In fact, what I mentioned in my review is AutoMix [2] in ECCV'2022 (open-source on GitHub), and I think it might be easy for you to compare with. It is also ok that you provide the comparison results based on AutoMix [1] in ECCV'2020 (a not open-source work).
> > > > > >
> > > > > > [1] Zhu, et al. Automix: Mixup networks for sample interpolation via cooperative barycenter learning. ECCV, 2020.
> > > > > >
> > > > > > [2] Liu, et al. AutoMix: Unveiling the Power of Mixup for Stronger Classifiers. ECCV, 2022.

---

> > > > > > > ### Author Response · Authors · 2022-08-08
> > > > > > > **Response to the Confusion of Citation**
> > > > > > >
> > > > > > > Hi Reviewer 13Sq,
> > > > > > >
> > > > > > > Thanks for pointing out this issue. We are sorry about the confusion. We indeed compared with AutoMix [2] (ECCV'2022) in our additional experiments. We made a mistake when adding the citation and have fixed this issue in the updated version. Many thanks!

---

### Official Review · Reviewer_t9o8 · 2022-07-15

**Rating:** 6
**Confidence:** 3
**Soundness:** 3 good
**Presentation:** 3 good
**Contribution:** 3 good

**Summary:**

This paper proposes a simple method for regression tasks. It improves the mixup data augmentation by selectively interpolating examples with similar labels, or re-weighting the probability of mixup pairs. Extensive experiments on various data modalities have been conducted to validate its effectiveness.

**Questions:**

1. lines 50-56 are hard to be understood. Does Figure 1(a) only present the ShapeNet1D Pose Prediction task? At the same time, why do pair 1 and pair 3 have close input similarities? Besides, why is the regression task more sensitive to noise than the classification task? Is there any literature to support this claim, and what's noise in the whole paper (Can an image have noise?)?

2. The choice of bandwidth. I first appreciate the ablation study of bandwidth sigma in section 4.5, but it has not convinced me of the claims that mixReg reduces the efforts to tune the bandwidth. For example, the label of ShapeNet1D has a range of [0, 360], and the other datasets may have a much different value range. How to deal with this problem to choose the correct sigma? Why is the output normalized to [0, 10] in line 270? The xlabels of Figure 3 are incorrect.

3. An essential property of mixup is the robustness of label noise. As mixReg uses the distances between labels to select the pairs, will the label noise harm the generalization of mixReg?

4. Can mixReg be applied to more complex and real-world natural tasks (eg, counting or pose estimation)? Mixup has shown its effectiveness on various classification tasks such as ImageNet. This paper only presents the experiments on some simple datasets, such as Shape1D and etc.

5. Will the code be publicly available?

**Ethics Review Area:**

["I don’t know"]

**Limitations:**

See above.

**Strengths And Weaknesses:**

### Strengths
* clear motivation and good presetation;
* Extensitive expermients;

### Weaknesses
I have not found the critical drawbacks of this paper. However, I have not carefully checked the correctness of sections 3.2, 3.3, and 3.4. In my opinion, section 3.1 has thoroughly presented the proposed method, and the rest provides little information but harmed the readability.

---

> ### Author Response · Authors · 2022-08-02
> **Response to Reviewer t9o8 (1/2)**
>
> Thank you for your valuable feedback to help us improve our paper. We have revised our paper based on your feedback. We detail our response below and please kindly let us know if our response addresses your concerns.
>
> > **Q1**: lines 50-56 are hard to be understood. Does Figure 1(a) only present the ShapeNet1D Pose Prediction task? At the same time, why do pair 1 and pair 3 have close input similarities?
>
> **A1**: Figure 1(a) only presents the ShapeNet1D Pose prediction task. We calculate the input feature similarities of pair 1, pair 2, and pair 3 as $1.51\times 10^5$, $1.82\times 10^5$, and $1.50\times 10^5$, respectively, where the results indicate that pairs 1 and 3 have close input similarities. We have revised the caption of Figure 1 to clarify it.
>
> ---
>
> > **Q2**: Besides, why is the regression task more sensitive to noise than the classification task? Is there any literature to support this claim, and what's noise in the whole paper (Can an image have noise?)?
>
> **A2**: We agree that this claim is based on our intuition. Intuitively, labels in classification are discrete and there are margins between classes, where subtle feature noises may not result in the change of labels. While labels in regression tasks are continuous, feature noises are more likely to lead to label changes. We have revised our paper and removed this sentence to make the introduction more precise, which does not affect our motivation and conclusion.
>
> ---
>
> > **Q3**: The choice of bandwidth. I first appreciate the ablation study of bandwidth sigma in section 4.5, but it has not convinced me of the claims that mixReg reduces the efforts to tune the bandwidth. For example, the label of ShapeNet1D has a range of [0, 360], and the other datasets may have a much different value range. How to deal with this problem to choose the correct sigma?
>
> **A3**: Thanks for pointing it out. We have revised our claim to make it more precise: mixReg reduces the efforts to tune the bandwidth $\sigma$ for every specific dataset. Roughly tuning the bandwidth in the range [0.01, 0.1, 1, 10, 100] is sufficient to get a relatively satisfying performance. In our experiments, we perform a grid search and apply cross-validation to find the best value of bandwidth. We have added an empirical discussion about how to pick bandwidth in Appendix F.3.1 of the revised paper.
>
> ---
>
>
> > **Q4**: Why is the output normalized to [0, 10] in line 270?
>
> **A4** In Pascal3D, as mentioned in Appendix D.1 of the initial submission, we follow [Yin et al. ICLR 2020] to preprocess the data, where the labels are normalized to [0, 10]. We have clarified it in Line 264-265 in the revised paper.
>
> ---
>
>
> > **Q5**: The xlabels of Figure 3 are incorrect.
>
> **A5**: Thanks for pointing this out. We have fixed the typo and updated Figure 3 in the revised paper.
>
> ---
>
> > **Q6**: An essential property of mixup is the robustness of label noise. As mixReg uses the distances between labels to select the pairs, will the label noise harm the generalization of mixReg?
>
> **A6**: We conduct experiments to investigate the robustness of mixReg to label noise, which is also added in Appendix F.4 of the revised paper. Specifically, we inject Gaussian noises into the labels. For each dataset, the noise is set as 30% of the standard deviation among the corresponding original labels, where adding noise significantly degrades the performance compared with clean data. In Table R1, we report the results and the corresponding noise distributions on Exchange-Rate, ShapeNet1D, and DTI, respectively.
>
> **Table R1**: Robustness analysis to label noise. Here, $\downarrow$ denotes the smaller the better; $\uparrow$ denotes the larger the better.
>
> | Model           | Exchange-Rate | ShapeNet1D   | DTI |
> |---------------|---------------|-------|------------|
> |              |  RMSE $\downarrow$ |  MSE $\downarrow$  |  Avg. R $\uparrow$ |
> | Noise Type | $\mathcal{N}(0, 1.18\times10^{-3})$ | $\mathcal{N}(0, 0.874)$ | $\mathcal{N}(0, 7.59\times10^{-3})$ |
> | ERM/MAML | 0.0381 $\pm$ 0.0014        | 5.553 $\pm$ 0.098      | 0.334 $\pm$ 0.018 |
> |  mixup/MetaMix   | 0.0375 $\pm$ 0.0017        | 5.329 $\pm$ 0.101      | 0.307 $\pm$ 0.021 |
> | **mixReg (ours)**                      | **0.0360 $\pm$ 0.0013**        | **5.185 $\pm$ 0.096**      | **0.356 $\pm$ 0.013** |
>
>
> According to Table R1, with the addition of label noise, we observe that mixReg still improves the performance over ERM and vanilla mixup, showing its effectiveness and robustness to label noise.

---

> > ### Author Response · Authors · 2022-08-02
> > **Response to Reviewer t9o8 (2/2)**
> >
> > > **Q7**: Can mixReg be applied to more complex and real-world natural tasks (eg, counting or pose estimation)? Mixup has shown its effectiveness on various classification tasks such as ImageNet. This paper only presents the experiments on some simple datasets, such as Shape1D and etc.
> >
> > **A7**: The paper submission already evaluates mixReg on many large-scale real-world datasets, including Echocardiogram Video (Echo), PovertyMap, and Drug-target Interaction (DTI). Concretely, Echocardiogram Video includes 10,030 apical-4-chamber echocardiography videos and is one of the largest public labeled medical video datasets for cardiac function assessments [Ouyang et al., Nature 2020]. PovertyMap is a regression dataset in the WILDS benchmark [Koh et al. ICML 2021], which captures the real-world distribution shifts [Yeh et al., Nature Communication 2020]. Drug-target Interaction is the only domain shift dataset in Therapeutics Data Commons [Huang et al., NeurIPS 2021], modeling the real-world drug-target interactions. We’ve revised our introduction and highlighted these datasets in Line 72-75 of the introduction (revised version).
> >
> > ---
> >
> > > **Q8**: Will the code be publicly available?
> >
> > **A8**: Yes, we will release the code upon publication.
> >
> > ---
> >
> >
> > **Reference**
> >
> > [Yin et al. ICLR 2020] Yin, Mingzhang, George Tucker, Mingyuan Zhou, Sergey Levine, and Chelsea Finn. "Meta-learning without memorization." ICLR 2020.
> >
> > [Ouyang et al., Nature 2020] Ouyang, David, Bryan He, Amirata Ghorbani, Neal Yuan, Joseph Ebinger, Curtis P. Langlotz, Paul A. Heidenreich et al. "Video-based AI for beat-to-beat assessment of cardiac function." Nature 580, no. 7802 (2020): 252-256.
> >
> > [Yeh et al. Nature Communication 2020] Yeh, Christopher, Anthony Perez, Anne Driscoll, George Azzari, Zhongyi Tang, David Lobell, Stefano Ermon, and Marshall Burke. "Using publicly available satellite imagery and deep learning to understand economic well-being in Africa." Nature communications 11, no. 1 (2020): 1-11.
> >
> > [Koh et al. ICML 2021] Koh, Pang Wei, Shiori Sagawa, Henrik Marklund, Sang Michael Xie, Marvin Zhang, Akshay Balsubramani, Weihua Hu et al. "Wilds: A benchmark of in-the-wild distribution shifts." In International Conference on Machine Learning, pp. 5637-5664. PMLR, 2021.
> >
> > [Huang et al. NeurIPS 2021] Huang, Kexin, Tianfan Fu, Wenhao Gao, Yue Zhao, Yusuf Roohani, Jure Leskovec, Connor W. Coley, Cao Xiao, Jimeng Sun, and Marinka Zitnik. "Therapeutics Data Commons: Machine Learning Datasets and Tasks for Drug Discovery and Development." In NeurIPS Datasets and Benchmarks. 2021.

---

> > > ### Comment · Reviewer_t9o8 · 2022-08-03
> > > **Response to  Authors**
> > >
> > > Thanks for the rebuttal, which has addressed most of my questions. I'm happy to see the manuscript can be improved by my comments.

---

> > > > ### Author Response · Authors · 2022-08-03
> > > > **Response to Reviewer t9o8**
> > > >
> > > > Dear Reviewer t9o8,
> > > >
> > > > Thank you for your quick response, we are happy to see that our response address most of your concerns. Thanks again for your valuable comments to help us improve our paper.

---

### Author Response · Authors · 2022-08-02
**Summary of Paper Revision**

We sincerely appreciate all reviewers for their insightful and constructive feedback. According to these comments, we have improved the paper (new pdf uploaded) and highlighted the main changes with blue text. Below, we summarize all changes:

1. Revised the caption of Figure 1 (Reviewer t9o8)

2. Detailed the choice of the bandwidth $\sigma$ and provided some empirical guidance in Appendix F.3.1 (Reviewers t9o8 and 13Sq)

3. Fixed typos in Figure 3 (Reviewer t9o8)

4. Added new experimental results and the discussion of mixReg with label noise in Section 4.4 (main paper) and Appendix F.4 (Reviewer t9o8)

5. Provided new experimental results and the discussion of the compatibility of mixReg with prior methods in analysis I of Section 4.4 (main paper) and Appendix F.1 (Reviewers 13Sq and VT7i)

6. Added more discussions about the comparison between mixReg and mixup and its variants in Appendix A.4 and Section 5 Related Work (Reviewer VT7i).

7. Revised some claims to make them more precise (Reviewers t9o8, 13Sq, VT7i)

8. Added more discussions about limitations and future work in Appendix G (Reviewer VT7i)

9. Fixed other typos in both the main paper and Appendix.

10. Since more results and discussions are included in the main paper, we put all empirical analytical experiments in Section 4.4. Accordingly, some Appendix indexes have been changed. We further simplify some descriptions in Section 2 to make the main paper 9 pages.

---

### Meta-Review · Area_Chair_ohZD · 2022-08-25

**Recommendation:** Accept
**Confidence:** Certain

**Metareview:**

This is an interesting and technically solid paper. The reviews are very consistent as well.

**Award:**

No

---

### Decision · Program_Chairs · 2022-09-14

Accept